EMBO
Molecular Medicine

# Altered metabolic landscape in IDH-mutant gliomas affects phospholipid, energy, and oxidative stress pathways

Fred Fack[1,†], Saverio Tardito[2,†], Guillaume Hochart[3,†], Anais Oudin[1], Liang Zheng[2], Sabrina Fritah[1], Anna Golebiewska[1], Petr V Nazarov[4], Amandine Bernard[1], Ann-Christin Hau[1], Olivier Keunen[1], William Leenders[5], Morten Lund-Johansen[6,7], Jonathan Stauber[3], Eyal Gottlieb[2], Rolf Bjerkvig[1,7] & Simone P Niclou[1,7,*] (iD)

## Abstract

Heterozygous mutations in NADP-dependent isocitrate dehydrogenases (IDH) define the large majority of diffuse gliomas and are associated with hypermethylation of DNA and chromatin. The metabolic dysregulations imposed by these mutations, whether dependent or not on the oncometabolite D-2-hydroxyglutarate (D2HG), are less well understood. Here, we applied mass spectrometry imaging on intracranial patient-derived xenografts of IDH-mutant versus IDH wild-type glioma to profile the distribution of metabolites at high anatomical resolution *in situ*. This approach was complemented by *in vivo* tracing of labeled nutrients followed by liquid chromatography–mass spectrometry (LC-MS) analysis. Selected metabolites were verified on clinical specimen. Our data identify remarkable differences in the phospholipid composition of gliomas harboring the IDH1 mutation. Moreover, we show that these tumors are characterized by reduced glucose turnover and a lower energy potential, correlating with their reduced aggressivity. Despite these differences, our data also show that D2HG overproduction does not result in a global aberration of the central carbon metabolism, indicating strong adaptive mechanisms at hand. Intriguingly, D2HG shows no quantitatively important glucose-derived label in IDH-mutant tumors, which suggests that the synthesis of this oncometabolite may rely on alternative carbon sources. Despite a reduction in NADPH, glutathione levels are maintained. We found that genes coding for key enzymes in *de novo* glutathione synthesis are highly expressed in IDH-mutant gliomas and the expression of *cystathionine-β-synthase* (*CBS*) correlates with patient survival in the oligodendroglial subtype. This study provides a detailed and clinically relevant insight into the *in vivo* metabolism of IDH1-mutant gliomas and points to novel metabolic vulnerabilities in these tumors.

**Keywords** CBS; glioma; isocitrate dehydrogenase; mass spectrometry imaging; phospholipids
**Subject Categories** Cancer; Metabolism; Neuroscience

## Introduction

Gliomas are a heterogeneous group of glial-derived brain tumors, the majority of which have a poor prognosis. The identification of heterozygous point mutations in the genes encoding for isocitrate dehydrogenase (IDH) 1 and 2 has been a breakthrough in the understanding of possible causes of glioma genesis (Parsons *et al*, 2008). IDH mutations are present in the vast majority of lower grade gliomas and define a subtype with a favorable prognosis (Hartmann *et al*, 2009; Yan *et al*, 2009). They have recently been recognized as the major determinant in the molecular classification of diffuse gliomas (Cancer Genome Atlas Research *et al*, 2015; Ceccarelli *et al*, 2016). Of the two isoforms associated with tumorigenesis, cytosolic IDH1 is the most commonly mutated, while the mitochondrial IDH2 isoform represents < 5% of mutant cases (Hartmann *et al*, 2009). Interestingly while IDH1 and IDH2 are NADP$^+$-dependent enzymes, the mitochondrial NAD-dependent IDH3 isoform has not been associated with mutations.

1 NorLux Neuro-Oncology Laboratory, Department of Oncology, Luxembourg Institute of Health, Luxembourg City, Luxembourg
2 Cancer Metabolism Research Unit, Cancer Research UK, Beatson Institute, Glasgow, UK
3 IMABIOTECH, Loos, France
4 Genomics and Proteomics Research Unit, Department of Oncology, Luxembourg Institute of Health, Luxembourg City, Luxembourg
5 Department of Pathology, Radboud University Medical Centre, Nijmegen, The Netherlands
6 Haukeland Hospital, University of Bergen, Bergen, Norway
7 Kristian Gerhard Jebsen Brain Tumor Research Center, Department of Biomedicine, University of Bergen, Bergen, Norway
*Corresponding author. Tel: +352 26970 273; Fax:+352 26970 390; E-mail: simone.niclou@lih.lu
†These authors contributed equally to this work

IDH enzymes are β-decarboxylating dehydrogenases that catalyze the reversible reaction of isocitrate to α-ketoglutarate (αKG, also known as 2-oxoglutarate), while generating NADPH and $CO_2$. The cancer-associated mutation of IDH (IDHm) has gained a neomorphic activity whereby αKG is converted to D-2-hydroxyglutarate (D2HG) while oxidizing NADPH. This reaction produces a 50- to 100-fold increase in D2HG levels in cells carrying the mutation (Dang *et al*, 2009). Established as an oncometabolite, D2HG has been shown to competitively inhibit TET methylcytosine dioxygenases, as well as JmjC domain-containing histone demethylases, thus resulting in global hypermethylation of DNA and chromatin (Xu *et al*, 2011). Recent data confirmed that IDHm is at the origin of epigenetic instability in glioma cells, leading to hypermethylation of CpG islands (G-CIMP phenotype) and histones H3 and H4 (Duncan *et al*, 2012; Turcan *et al*, 2012).

Next to these epigenetic alterations, these tumors are likely to harbor D2HG-independent metabolic abnormalities linked to the loss of the normal function of IDH, which plays a major role in, for example, the control of mitochondrial redox balance and cellular defense against oxidative damage (Jo *et al*, 2001; Lee *et al*, 2002). The oncogenic adaptations at the metabolic level are, however, incompletely understood because of a lack of appropriate cell culture and animal models. Glioma cells carrying an endogenous IDH mutation are notoriously difficult to grow in culture or to expand *in vivo* in rodent models, and if so, they grow very slowly. Only a handful of patient-derived xenografts (PDX), including ours, have been described in the literature (Luchman *et al*, 2012; Klink *et al*, 2013; Navis *et al*, 2013). While glioblastoma (GBM) cell lines engineered to overexpress the mutant enzyme have served as useful models for metabolic profiling (Reitman *et al*, 2011; Mohrenz *et al*, 2013; Ohka *et al*, 2014; Shi *et al*, 2014), this set up harbors some limitations: The mutation is introduced out of the appropriate genetic background, the stoichiometric balance of the mutant versus wild-type form (IDHwt) is disrupted, and the specificities of the brain microenvironment, likely to impact cellular metabolism, are not maintained. Metabolite profiling is further hampered by intrinsic limitations of available analytical technologies in as such that no single methodology can provide a comprehensive metabolic landscape.

In an attempt to shed light on the mechanistic link between IDH mutations and oncometabolic adaptations, we profiled the *in vivo* metabolic content of patient-derived glioma xenografts and clinical glioma samples with or without the IDH mutation. Using mass spectrometry-based imaging (MSI) on brain sections, we provide an anatomical distribution of metabolites, complemented by liquid chromatography–mass spectrometric (LC-MS) analyses and *in vivo* metabolic tracer studies. Our data show major alterations in lipid metabolism, significant adaptations in the central energy metabolism, and oxidative stress pathways in IDHm gliomas, pointing to novel metabolic vulnerabilities that may be therapeutically exploitable.

## Results

### Altered lipid metabolism in IDH1-mutant glioma xenografts

We have previously generated intracranial patient-derived xenografts (PDXs) of glioblastoma (GBM) and have shown that such tumors recapitulate human GBM growth patterns, retaining phenotypic and genetic aberrations of the parental tumors (Fack *et al*, 2015; Bougnaud *et al*, 2016). In comparison, PDXs of lower grade diffuse gliomas carrying the IDH mutation have long been difficult to establish, and no cell culture models of these glioma subtypes exist. Recently, we and others have been able to implant such PDX and maintain them by serial transplantation in the brain of immunodeficient mice (Luchman *et al*, 2012; Klink *et al*, 2013; Navis *et al*, 2013). IDHm glioma PDXs display IDH1m immunoreactivity and a more than 50-fold increase in D2HG levels in the tumor area compared to IDHwt PDXs (Navis *et al*, 2013) and in line with their clinical behavior develop very slowly in the mouse brain (between 2–10 months). Nevertheless, such experimental models allow for the first time a comparative metabolomic analysis of different glioma subtypes in the brain microenvironment. To this aim, we profiled brain sections of three IDHm and three IDHwt glioma xenografts by MSI to generate *in situ* distribution maps of tumor metabolites (Fig 1A). This strategy comprised a large-scale untargeted analysis performed on small regions of interest (ROI) and a targeted approach to quantify the distribution of selected metabolites in a large ROI (Fig 1A). Comparison to control mouse brain tissue (CB) served to calibrate the data for reliable comparison between tumors. IDHwt PDXs (P3, T434, P8) were GBM-derived, while IDHm PDXs contained two lower grade gliomas (LGG: E478, T186) and one glioblastoma (T394) (see diagnostic details in Table 1). It should be noted that the new WHO classification for brain tumors (Louis *et al*, 2016) has introduced the IDH status as a defining feature of glioma subtype and largely separates lower grade diffuse gliomas (IDHm) from grade IV glioblastoma (IDHwt). The inclusion of a relatively rare IDHm GBM should allow to distinguish metabolic effects inherent to IDH status or linked to tumor grade.

To get an unbiased overview of major differences, we first performed untargeted profiling on three samples (IDHm and IDHwt PDX and normal control brain (CB)) revealing more than 100 metabolites unevenly distributed between IDHm versus IDHwt gliomas (Tables EV1–EV3). Interestingly, many of the top metabolites had a relatively high mass range (as indicated by their mass-to-charge ratio ($m/z$) > 600), often representing metabolites of unknown identity. Although not all mass ratios could be attributed to a specific metabolite, the majority was tentatively assigned using Metlin, HMBD or an internal metabolite database. To those entities matching multiple isobaric compounds, several metabolite identities were assigned (Tables EV2 and EV3). Striking differences associated with IDH status were found in phospholipid compounds, including phosphatidylethanolamine (PE), phosphatidylinositol (PI), phosphatidylserine (PS), phosphatidylcholine (PC), and phosphoglycerol (PG), indicating major aberrations in lipid composition of IDHm tumors. Using the targeted MSI approach, we verified the detection of selected metabolites in all six PDX samples. Figure 1B depicts the distribution of D2HG in IDHm PDX allowing to clearly identify the tumor region on brain sections. D2HG levels were very high in all IDHm PDXs (at least 50-fold), as confirmed by LC-MS (Fig EV1A). Several higher mass metabolites (e.g., $m/z$ 778.47; $m/z$ 807.53) were accumulated in IDHm gliomas, with much lower detection in IDHwt tumors (Fig 1B and C; MSI images in Fig EV1B). Some of them were also present in fiber tracts of normal brain areas (see, e.g., $m/z$ 778.47 in Fig EV1B). Interestingly, two putative phospholipids ($m/z$ ratio

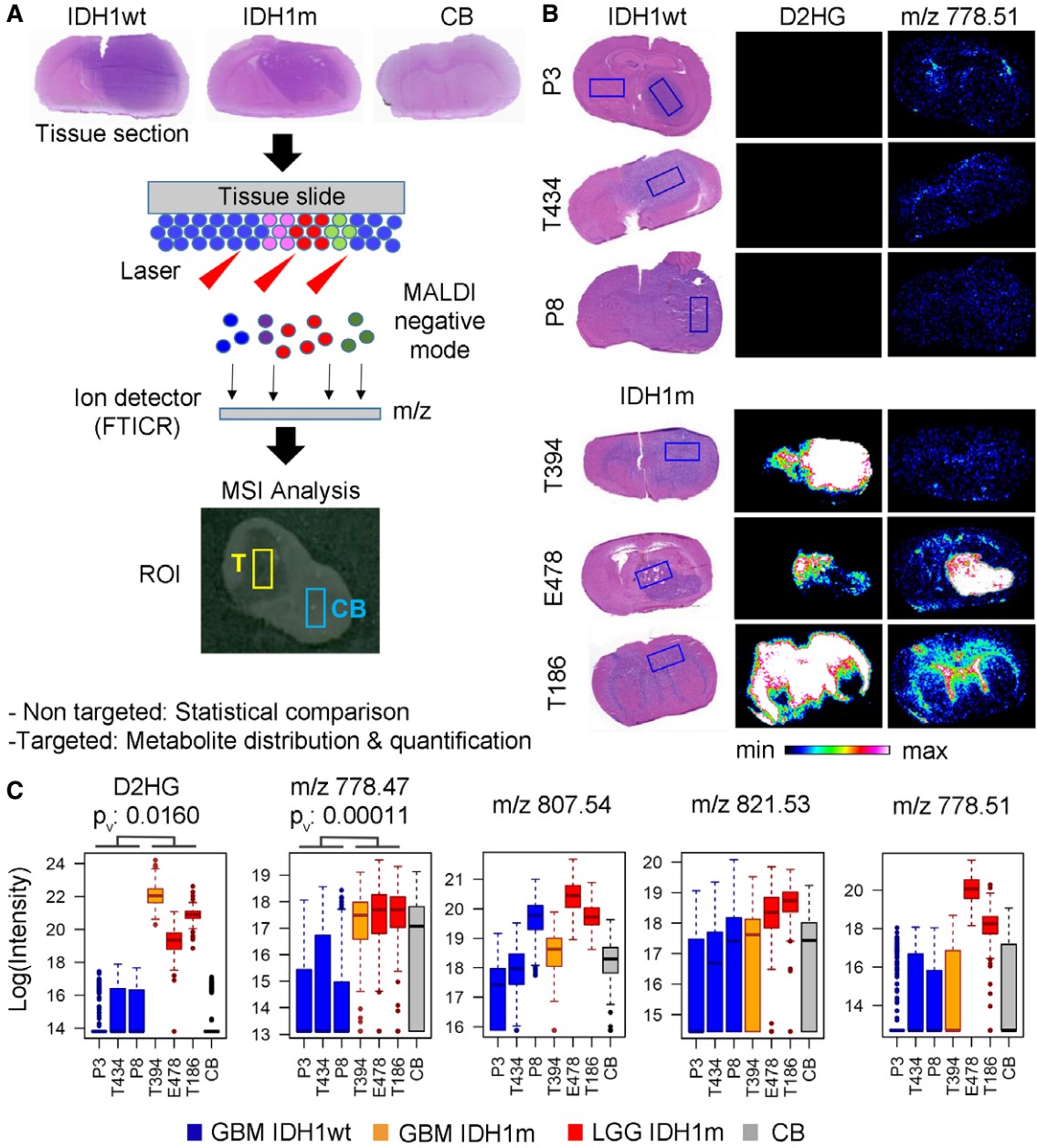

**Figure 1.  MALDI imaging for *in situ* metabolic profiling of gliomas reveals aberrant phospholipid metabolism in IDH-mutant glioma.**

A   MALDI imaging was performed on tumor-containing brain sections at a lateral resolution of 100 μm, and (M-H) ions were analyzed in negative mode with a FTICR mass spectrometer (analytical range 100–1,000 Da). Hematoxylin–eosin (HE) staining of the same sections was performed after MALDI analysis. Patient-derived glioma xenografts (PDXs) with IDH1 wild type (IDH1wt), IDH1 mutant (IDH1m), and control brain tissue (CB) were analyzed in a non-targeted and targeted approach using regions of interest (ROI) of different sizes to compare tumor (T, yellow line) and contralateral control brain (CB, blue line). The non-targeted approach was used for statistical pair-wise comparisons between samples, while the targeted approach on a selected set of metabolites allowed distribution and quantification analyses.

B   Histological sections of three IDH1wt PDXs (P3, T434, and P8) and three IDH1m PDXs (T394, E478, and T186) showing a large ROI (boxed rectangle, > 500 pixels) within the tumor area applied for targeted quantification of selected metabolites. Left rectangle in P3 PDX was used as control contralateral brain (CB) for quantifications. Middle panel shows tissue distribution of D-2-hydroxyglutarate (D2HG) by MSI, which is exclusively detected in IDH1m PDXs (D2HG in IDH1wt tumors is below detection limit). Right panel shows tissue distribution of a key metabolite (*m/z* 778.51) presenting strong accumulation in IDH1m lower grade gliomas (LGG) versus glioblastomas (GBM) independent of IDH status. An intensity-dependent color code indicates the relative amount of a specific compound (defined by *m/z* value) throughout the tissue section.

C   Quantification of several high mass metabolites differentially present in IDH1m PDXs (IDH1wt glioblastomas (GBM) in blue, IDH1m GBM in orange, IDH1m lower grade gliomas (LGG) in red, and contralateral control brain (CB) in gray). Box plots represent log values of metabolite intensities measured within a large ROI (> 500 pixels). Box limits indicate the 25th and 75th percentiles and center lines show the medians as determined by R software; whiskers represent the extreme low and high observed values, unless those are above 1.5 times interquartile range (IQR) – thereby whiskers are limited to 1.5 IQR. All outlying data points are represented by dots. Many metabolites in this mass range (*m/z* 700–900) have been putatively assigned to phospholipids (Tables EV2 and EV3). *m/z* represents the mass over charge ratio of ionized metabolites, as measured by the mass spectrometer. The statistical significance was calculated for IDH1m (*n* = 3) versus IDH1wt (*n* = 3) PDXs using *t*-test. $P_v$: *P*-values.

**Table 1.    Neuropathological diagnostics and genetic characterization of samples used for patient-derived xenografts.**

| Xenografts | Histopathological diagnosis | Molecular pathology (methylation score) | Survival (days) | IDH status | 1p/19q Codel | ATRX mutation | CDKN2A/B deletion | RTK amplified | Gain of Chr7 | Loss of Chr10 |
|---|---|---|---|---|---|---|---|---|---|---|
| E478 | Oligodendroglioma grade III | NA | 73 days | IDH1-R132H[b] | Yes[a] | NA | No[a] | No[a] | Yes[a] | Yes[a] |
| T186 | Anaplastic oligodendroglioma grade III | NA | 107 days | IDH1-R132H[b] | No[a] | Yes[b] (E1473) | No[a] | No[a] | No[a] | Partial[a] |
| T394 | Glioblastoma grade IV | Glioma, IDH mutant, subclass high-grade astrocytoma (0.99) | 77 days | IDH1-R132H[b] | Partial[a] | Unclear significance[b,c] | Yes[a] | PDGFRA[a] | No[a] | Partial[a] |
| P3 | Glioblastoma grade IV | NA | 40 days | No[b] | No[a] | No[b] | Yes[a] | No[a] | Yes[a] | Partial[a] |
| P8 | Glioblastoma grade IV | NA | 58 days | No[b] (V178I) | No[a] | Unclear significance[b,d] | Yes[a] | EGFR[a] | Yes[a] | Yes[a] |
| T434 | Glioblastoma grade IV | Glioblastoma, IDH wild-type (0.99) | 42 days | No[b] | No[a] | No[b] | No[a] | EGFR[a] | Partial[a] | Yes[a] |

NA: not assessed; ATRX E1473: Nonsense mutation, SIFT & Polyphen-2 score deleterious.

IDH1 V178I − no functional domain affected, 2-HG unaffected.

[a]Array comparative genomic hybridization.

[b]Targeted next generation sequencing.

[c]ATRX Q891E − hypomorphic mutation, protein features might be affected; additional 15 missense mutations with unclear significance.

[d]ATRX Q891E − hypomorphic mutation, protein features might be affected; additional S502C missense mutation with unclear significance.

of 778.51 and *m/z* 821.53) appeared specifically enriched in lower grade IDHm PDX compared to GBMs (Fig 1C; *m/z* 778.51 more than 1,000-fold increase, Table EV2), thus correlating with grade rather than IDH status. Similar to D2HG, IDHm-specific metabolites were restricted to the tumor area, with limited diffusion/ extension toward adjacent brain areas, yet MSI allowed to detect intra-tumoral metabolic heterogeneity (Fig 1B). In summary, these data reveal drastic alterations in the composition of phospholipid and other high mass metabolites, some of them being related to tumor grade. Although the unequivocal identification of these metabolite species remains to be determined, they may represent putative novel biomarkers for IDH-mutant gliomas.

### IDH-mutant glioma display reduced energy potential

Targeted MSI was further focused on phospho-metabolites related to cellular energy pathways. Hexose phosphates (Hex-P) showed increased levels in all tumors compared to normal brain, indicative of enhanced glucose uptake (Fig 2A, quantification in Fig EV1B). Similarly, adenosine diphosphate (ADP) and adenosine triphosphate (ATP) nucleotides were increased in tumors compared to the brain (Fig 2A, quantification Fig EV1B). However, while adenosine monophosphate (AMP) and ADP levels were similar in all tumors, ATP levels were lower in IDHm compared to IDHwt PDX, as quantified by LC-MS (Fig 2B). As a result, the overall energetic charge index (defined as ATP+0.5ADP/(ATP+ADP+AMP)) was significantly lower in IDHm gliomas (Fig 2C and D). This correlated with the generally lower proliferation index of IDHm xenografts (e.g., MIB

index based on Ki67 staining was $21.8 \pm 3\%$ for E478 and $54.7 \pm 2\%$ for P3 PDX) and the slower tumor development in mice (Table 1), which is also in agreement with the better prognosis of IDHm glioma patients observed in the clinic. Similar to D2HG, the energy charge distribution displayed a gradient from the tumor core to the periphery (Fig 2C), which may reflect a reduced ratio of cancer versus non-neoplastic cells. Sedoheptulose-7-P (Sdh7-P), an intermediate of the pentose phosphate pathway and a precursor of ribose phosphate, was also lower versus high grade gliomas (Fig 2B), in line with a reduced *de novo* biosynthesis of nucleotides. In summary, these data indicate a reduced energy potential in IDHm gliomas that may explain the relatively low proliferation rate of this subtype of brain tumors.

### Metabolic flux toward αKG and D2HG is unresolved in IDHm gliomas

In order to understand the origin of reduced energy potential in IDHm gliomas, we applied LC-MS to quantify steady-state level of glucose-derived metabolites in the six PDXs and performed *in vivo* tracing of isotope-labeled metabolites after injection of $^{13}C_6$-glucose in two PDXs. There was a tendency toward reduced levels of glucose-derived metabolites (e.g., pyruvate, lactate, citrate) in IDHm tumors, which reached significance for the tricarboxylic acid (TCA) cycle intermediates αKG and malate (Fig 3A). A similar tendency was also observed in a set of clinical glioma samples analyzed by LC-MS (7 IDHwt versus 6 IDHm), where a significant drop in pyruvate was noted (Fig EV4). Of note, the metabolite levels of the

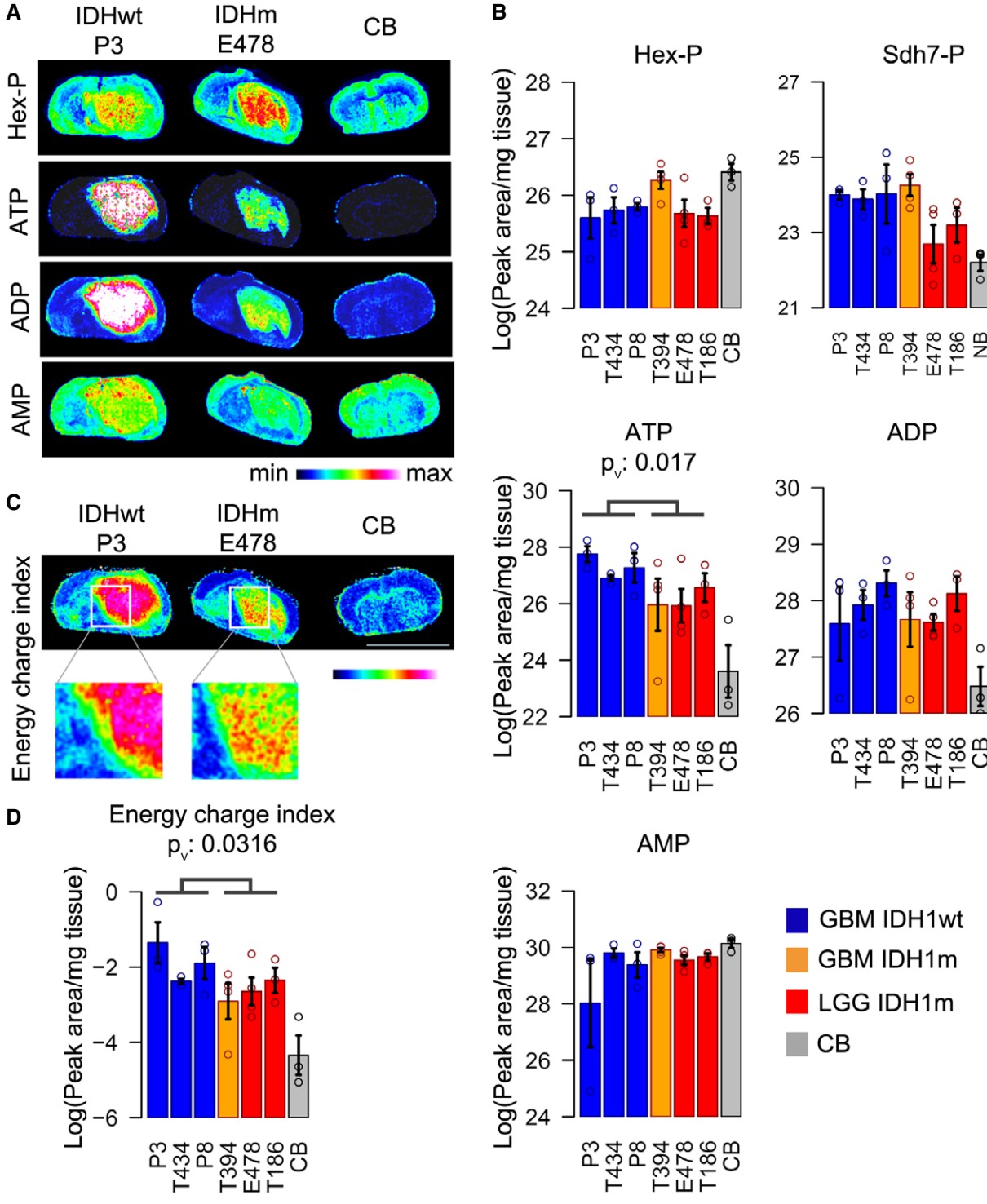

**Figure 2. IDH-mutant gliomas display a low energy potential.**

A MALDI imaging of indicated metabolites in IDHwt PDX (P3), IDHm PDX (E478), and control brain (CB). Hex-P: hexose phosphate; AMP, ADP, ATP: adenosine mono-, di-, and triphosphate, respectively.

B LC-MS quantification of indicated metabolites in six PDXs. Relative quantities of metabolites are expressed in log scale and normalized to tissue quantity used for extract preparation. IDHwt glioblastomas (GBM) in blue, IDHm GBM in orange, IDHm lower grade gliomas (LGG) in red, contralateral control brain (CB) in gray. Sdh7-P: sedoheptulose-7-phosphate (*n* = 3/sample/group).

C Energetic charge defined as the ratio of (ATP+1/2ADP)/(ATP+ADP+AMP) was calculated at each pixel to present its distribution throughout the tumor. Scale bar: 8.2 mm.

D The energetic charge ratio was confirmed by the LC-MS data performed on separate tumor extracts (*n* = 3/sample/group).

Data information: The indicated statistically significant group differences are based on *t*-test, $P_v$: *P*-values. Error bars in (B and D) represent standard error of the mean.

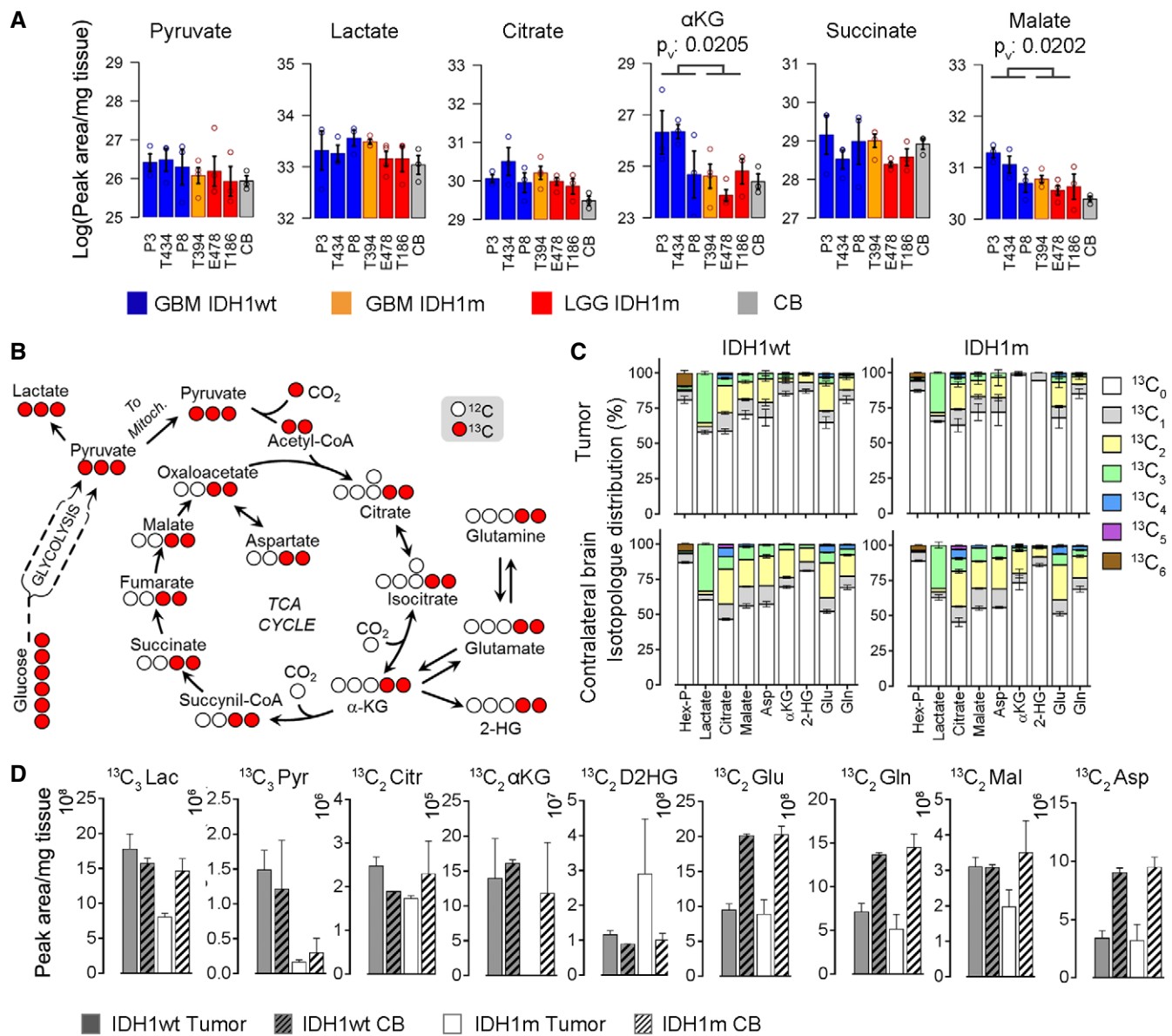

**Figure 3. *In vivo* metabolic flux analysis indicates a reduced glucose turnover in IDH-mutant glioma.**

A   LC-MS quantification of steady-state level of glycolysis and TCA cycle metabolites in three IDH1wt and three IDH1m PDXs and contralateral control brain (CB) ($n$ = 3/sample/group). Error bars represent standard error of the mean. Metabolites showing statistically significant group differences between IDH1wt and IDH1m are indicated ($P$-value < 0.05). Color code as in Figs 1 and 2. Indicated statistical significant group differences are based on $t$-test, $P_v$: $P$-values.

B   Schematic of $^{13}C_6$ glucose-derived carbon flux to pyruvate and TCA metabolites, highlighting expected $^{13}C_2$ and $^{13}C_3$ labels (in red) in intermediate metabolites.

C   *In vivo* flux analysis of $^{13}C$-labeled glucose injected 20 min prior to sacrifice in one IDH1wt PDX (P3) and one IDH1m PDX (E478) ($n$ = 3/sample). The percentage of glucose-derived $^{13}C$ label is shown in selected metabolites containing 0 to 6 $^{13}C$ atoms ($^{13}C_0$-$^{13}C_6$ as indicated by color code). Note the absence of $^{13}C_2$ label of αKG and D2HG in IDHm tumors, while the label was detectable in IDHwt PDX and in contralateral brain of both tumors (lower panel). Error bars represent standard error of the mean.

D   Quantities of $^{13}C_3$- and $^{13}C_2$-labeled metabolites from glucose tracer experiment in (C). D2HG in IDH1m PDX shows a detectable $^{13}C_2$ signature here, but represents a minor fraction of the total cellular D2HG pool in this tumor. Hex-P: hexose phosphate, PEP: phosphoenolpyruvate, Pyr: pyruvate, Cit: citrate, Mal: malate, Asp: aspartate, Glu: glutamate, Gln: glutamine, α-KG: α-ketoglutarate, D2HG: D2-hydroxyglutarate. Error bars represent standard error of the mean.

IDHm GBM (in orange) appeared occasionally closer to the GBM group than to the IDHm lower grade tumors.

To gain a better understanding of the flux of glucose carbons in IDHm gliomas, $^{13}C_6$-glucose was injected in two PDXs (E478 and P3) 20 min before sacrifice (Fig 3B–D). We observed a reduction in $^{13}C$-labeled lactate, pyruvate and malate in IDHm tumors

compared to IDHwt, suggesting a reduced turnover of glucose metabolism (Fig 3D). Surprisingly, we did not detect heavy isotopologues of αKG and D2HG in IDHm gliomas (Fig 3C). This was in sharp contrast to IDHwt tumors and contralateral brain regions, where a significant fraction of $^{13}C_2$ isotopologues were present in both αKG and D2HG (Fig 3C and D). Although a small

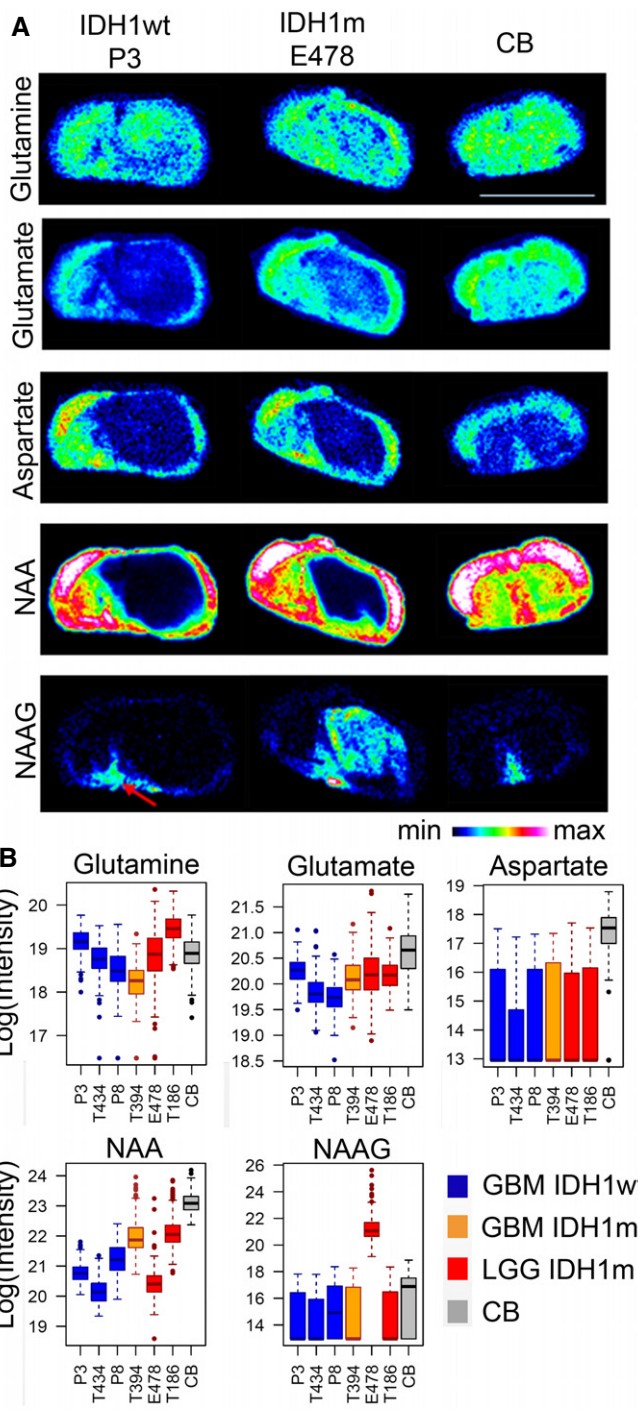

**Figure 4. Distribution of selected amino acids in gliomas.**

A MALDI imaging of key metabolites in IDH1wt (P3) and IDH1m (E478) glioma PDX, compared to control brain (CB) sections. Scale bar: 8.2 mm. Red arrow indicates brain region with high NAAG content.

B Quantification of metabolite distribution by MSI in six PDX with IDH1wt and IDH1m status (*n* = 3/group) and in contralateral control brain (CB). Box limits indicate the 25th and 75th percentiles and center lines show the medians as determined by R software; whiskers represent the extreme low and high observed values, unless those are above 1.5 times interquartile range (IQR) – thereby whiskers are limited to 1.5 IQR. All outlying data points are represented by dots. No significant differences between IDH groups were detected by *t*-test on the means, although significant differences are found between tumors and control brain (glutamate, aspartate). Color code as in Fig 1. NAA: N-acetylaspartic acid; NAAG: N-acetylaspartylglutamic acid.

Reliance on glutamine for anaplerosis implies the deamination of glutamine to glutamate by glutaminase in the mitochondrion, where glutamate can be further deaminated to αKG to replenish the TCA cycle. However, only a very small amount of $^{13}C_5$-glutamine was found in the tumors irrespective of their IDH status, and the heavy isotopologues of glutamine derivatives were not detectable, neither in control brain nor in tumors (Fig EV2). This is in agreement with our previous observation in GBM xenografts, confirming that in the mouse brain, glutamine is hardly taken up from the circulation (Tardito *et al*, 2015). It remains to be seen to what extent brain-derived glutamine or glutamate plays a role as bioenergetic substrates in IDHm gliomas. It should be noted, however, that based on the high level of glucose-derived glutamine and glutamate isotopologues detected in control brain (Fig 3C lower panel), it is surprising to see that this was not translated in $^{13}C_2$-labeling of αKG and D2HG in the tumor.

## Alterations in amino acids and neurotransmitters in IDH-mutant glioma

Glutamine is thought to be an important nutrient in cancer (DeBerardinis & Cheng, 2010), and IDHm tumors have been proposed to be particularly dependent on glutamine (Seltzer *et al*, 2010; Emadi *et al*, 2014), however, whether this holds true in the brain milieu with its distinct metabolite composition remains to be determined. In glioma PDXs, we found no relevant difference in glutamine and glutamate levels between IDHwt and IDHm tumors by MSI, although glutamate was reduced compared to control brain (Fig 4A and B), which was validated by LC-MS analysis (Fig EV3). LC-MS on clinical glioma samples confirmed similar glutamine levels in all gliomas, while a reduction in glutamate was detected in IDHm glioma patients (Fig EV4). Although this may suggest an increased reliance on glutamate rather than glutamine in these tumors, it is currently not clear why this observation was not reproduced in the PDX models.

Similar to glutamate, MSI showed a strong reduction in the excitatory amino acid neurotransmitter aspartate and its derivative N-acetyl-aspartate (NAA) in tumors compared to control brain, independent of IDH status (Fig 4A and B). NAA can be hydrolyzed by aspartoacylase (ASPA) to produce aspartate and acetate (Tsen *et al*, 2014) which was recently identified as a major bioenergetic substrate in brain tumors (Mashimo *et al*, 2014). NAA is also a substrate for the synthesis of N-acetylaspartylglutamate (NAAG), a

amount of $^{13}C_2$-labeled D2HG was seen in the IDHm tumor (Fig 3D), this fraction is negligible compared to total D2HG levels in this tumor. As the overall amount of αKG was comparably low in contralateral brain and IDHm tumor tissues (Fig 3A), the absence of labeling in the latter cannot simply be explained by falling below the detection limit. This suggested that mitochondrial IDH2 does not efficiently compensate glucose-derived αKG production in this setting and raised the question of the carbon source for αKG and D2HG in IDHm tumors. In an attempt to address this, we traced $^{13}C_5$-glutamine in IDHm and IDHwt PDXs.

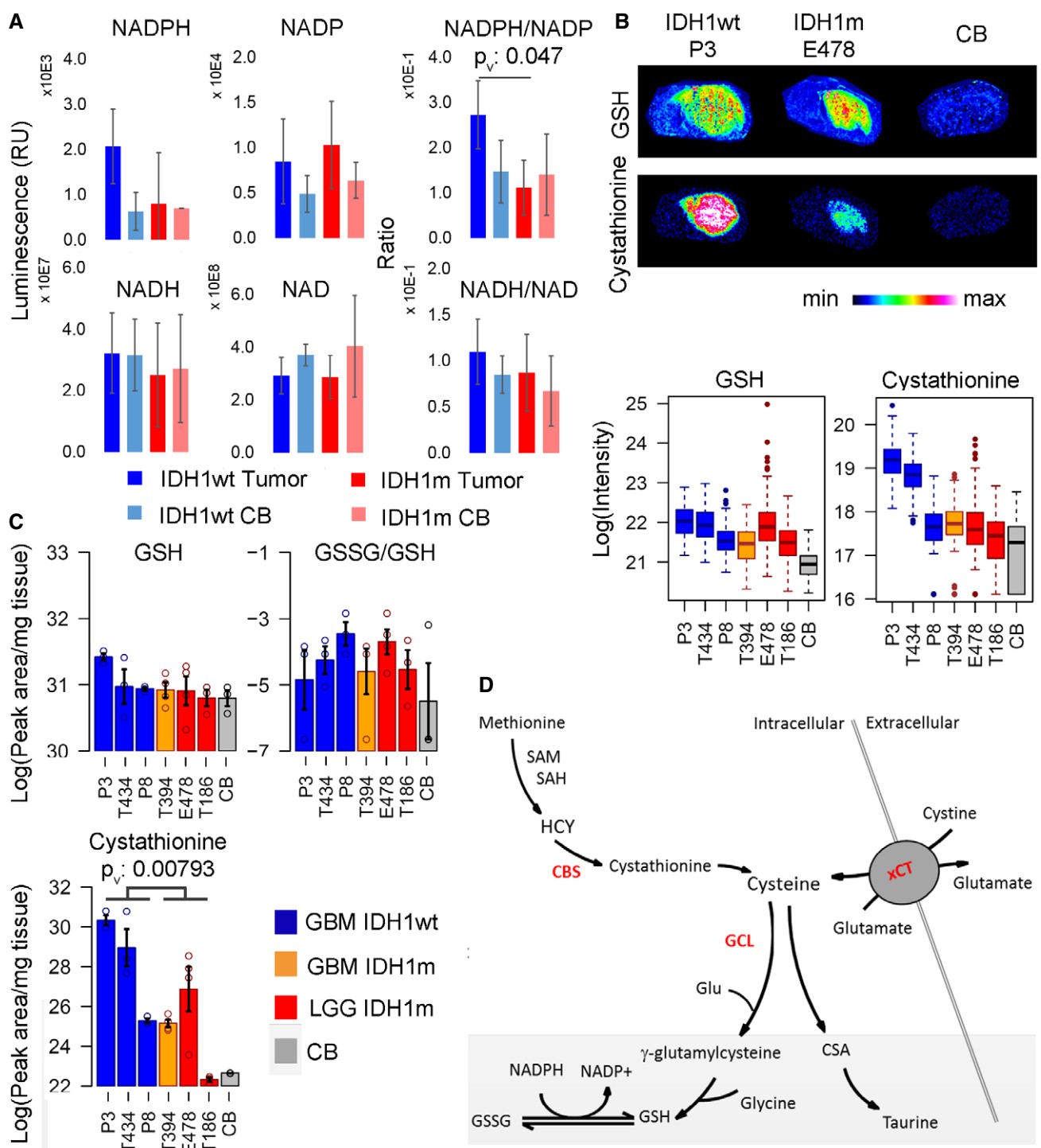

**Figure 5. Oxidative stress response in IDH wild-type and IDH-mutant glioma.**

A  NADPH levels are reduced in IDHm compared to wild-type tumors, resulting in a reduced NADPH/NAD+ ratio (n = 3/sample). Error bars represent standard deviation. *P*-value from *t*-test: < 0.05. NADPH, NADP: reduced, oxidized nicotinamide adenine dinucleotide phosphate, NADH, NAD: reduced, oxidized nicotinamide adenine dinucleotide.

B  MALDI images for glutathione (GSH) and cystathionine on IDH1wt (P3) and IDH1m (E478) PDXs. Quantification of MSI signals for GSH and cystathionine from 6 PDXs (n = 3/group). Box limits indicate the 25th and 75th percentiles and center lines show the medians as determined by R software; whiskers represent the extreme low and high observed values, unless those are above 1.5 times interquartile range (IQR) – thereby whiskers are limited to 1.5 IQR. All outlying data points are represented by dots.

C  LC-MS quantification of independent samples for GSH, GSSG (oxidized GSH), GSSG/GSH ratio and cystathionine from 6 PDXs (n = 3/sample/group). Statistical significance between IDH groups calculated by *t*-test, *P*$_v$: *P*-value. Error bars represent standard error of the mean.

D  Schematic of GSH synthesis, cysteine supply, and its degradation pathways. GSH: reduced glutathione; GSSG: oxidized glutathione; SAM: S-adenosylmethionine, SAH: S-adenosylhomocysteine, CSA: cysteine sulfinic acid. Key enzymes in red: CBS: cystathionine-beta-synthase, GCL: glutamate-cystine ligase, xCT: cysteine-glutamate antiporter.

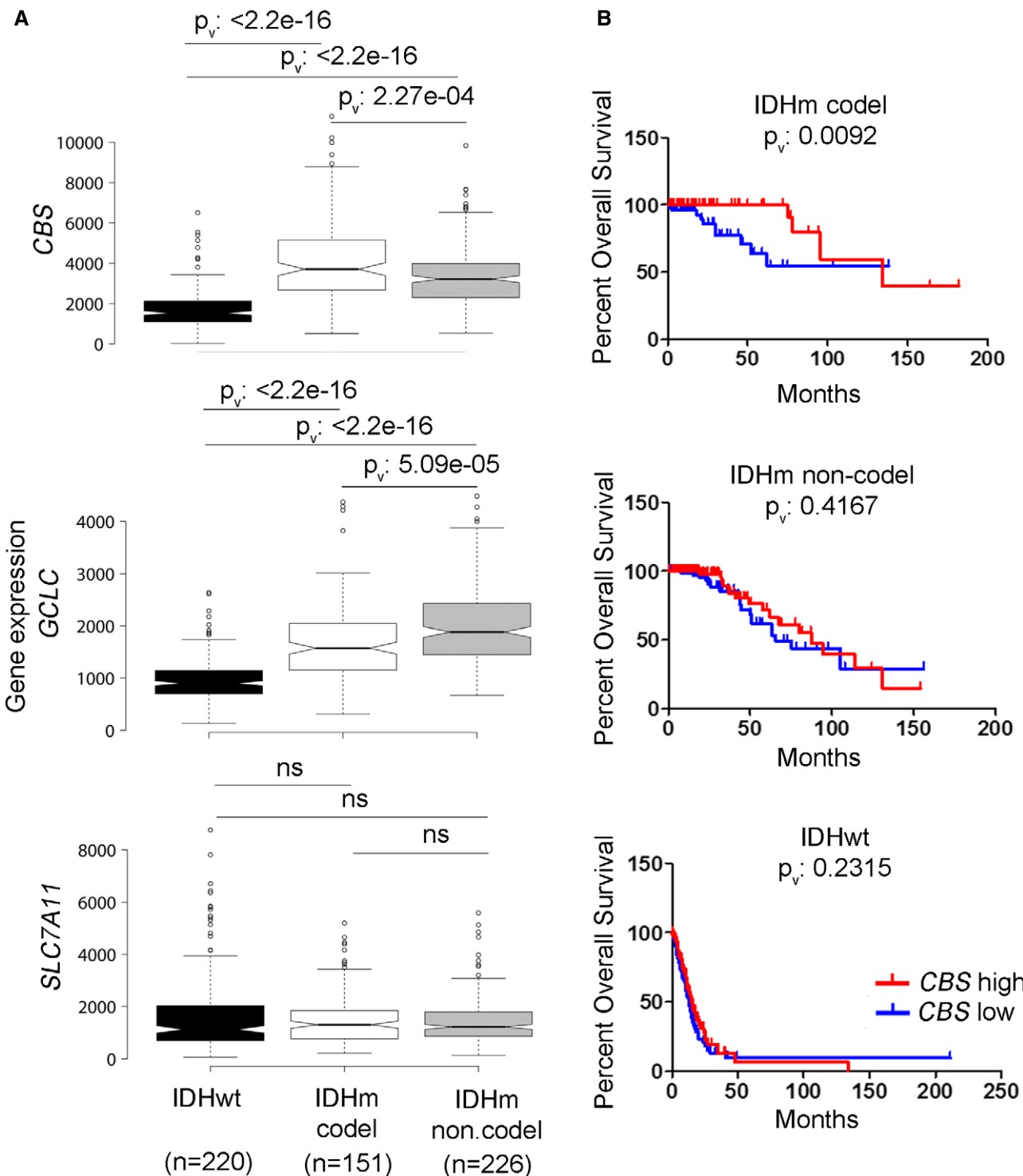

**Figure 6. *CBS* is a prognostic biomarker for patients with IDH-mutant 1p/19q co-deleted gliomas.**

A  Comparative gene expression analysis of key enzymes involved in GSH and cysteine synthesis and supply in IDHwt ($n$ = 220), IDHm with 1p/19q co-deletion ($n$ = 151), or IDHm without 1p/19q co-deletion ($n$ = 226) patients based on TCGA RNA-seq data from a cohort of 597 patients. CBS: Cystathionine-beta-synthase, key enzyme of the transsulfuration pathway. GCLC: catalytic subunit of glutamate-cysteine ligase. *SLC7A11*: gene encoding xCT, the cystine-glutamate antiporter. Center lines of the box plots show the medians; box limits indicate the 25th and 75th percentiles, outliers are represented by dots. Notches indicate ±1.58 × IQR/sqrt(n) and gives 95% confidence that two medians differ. Whiskers extend 1.5 times the interquartile range from the 25th and 75th percentile. Statistical test: Wilcoxon rank sum test with continuity correction. $P_v$: P-value, ns: non-significant.

B  Kaplan–Meier curves of CBS gene expression versus overall survival in patients with IDHwt, IDHm with or without 1p/19q co-deletion. Comparison of survival curves was done using Log-rank (Mantel-Cox) test.

neuropeptide abundant in specific brain areas, as seen by MSI of normal mouse brain (arrow in Fig 4A). While NAAG was not detectable in IDHwt PDXs, it showed high variability in IDHm tumors (Fig 4A). Identification of NAA and NAAG was validated by MALDI using isotopically labeled reference compounds (not shown), and quantification was confirmed by LC-MS (Fig EV3). Similar data were also found in clinical IDHm samples analyzed by LC-MS (Fig EV4), where two NAAG subpopulations were observed: Three of six displayed high levels of NAAG while the remaining three displayed levels similar to IDHwt samples, suggesting activation of different metabolic pathways independent of available biomarker status (IDH, 1p/19q, or methylation status; Table EV6).

### The antioxidant response in IDH-mutant gliomas

We next turned to address the oxidative state of IDHm gliomas. Based on the NADPH-dependent production of D2HG, it was proposed that these tumors display reduced NADPH levels (Bleeker *et al*, 2010), although direct NADPH quantification has not been provided. As a consequence, one would expect reduced glutathione (GSH) levels, the most abundant intracellular antioxidant protecting cells against free radicals (Townsend *et al*, 2003), because glutathione reductase reduces the disulfide bond of the oxidized form of GSH (glutathione disulfide: GSSG) in a NADPH-dependent reaction. We measured indeed a reduced NADPH/NADP$^+$ ratio in IDHm xenografts compared to IDHwt, an effect which was specific to NADPH, since the NADH/NAD$^+$ ratio was comparable between tumor types (Fig 5A). Interestingly, while MSI showed that the GSH level was much higher in tumors compared to normal brain, no significant difference was observed between IDHm and IDHwt tumors (Fig 5B). This was confirmed by LC-MS, which also allowed to measure GSSG. The GSSG/GSH ratios were similar in IDHm versus IDHwt PDXs, indicating that despite reduced NADPH, IDHm tumors appear to exhibit an adequate oxidative stress response (Fig 5C). Comparable GSH levels between mutant and wild-type gliomas were also found in the clinical specimen (Fig EV4).

These data suggested that GSH production may preferentially rely on *de novo* biosynthesis from cysteine. The transsulfuration pathway which supplies cysteine from methionine and cystathionine (Fig 5D) was recently shown to be a significant contributor to GSH synthesis in astrocytes (McBean, 2012). Although we were not able to directly measure cysteine (neither by MSI nor by LC-MS), we found lower levels of the cysteine precursor metabolite cystathionine in IDHm gliomas by MSI (Fig 5B), suggesting high turnover. Reduced cystathionine was confirmed by LC-MS (Fig 5C). We next interrogated the TCGA database for the expression of the key genes involved in GSH/cysteine metabolism: cystathionine-beta-synthase (*CBS*), the first enzyme of the transsulfuration pathway, γ-glutamylcysteine ligase catalytic subunit (*GCLc*) which catalyzes the ligation of cysteine and glutamate (Sikalidis *et al*, 2014), and the cystine-glutamate antiporter (xCT, *SLC7A11*), responsible for cysteine uptake (Fig 5D). The analysis was focused on IDHm versus IDHwt gliomas and further separated IDHm gliomas based on the co-deletion of chromosome 1p/19q to distinguish the oligodendroglial and astrocytic lineage. Based on the expression of almost 600 patients, we found that the levels of *CBS* and *GCL*c were significantly upregulated in IDHm compared to IDHwt gliomas (Fig 6A) supporting a higher

activation of the transsulfuration pathway. No significant difference was observed for xCT expression (*SLC7A11*) (Fig 6A). Among IDHm gliomas, *CBS* expression was higher in IDHm-1p/19q co-deleted gliomas compared to IDHm without 1p/19q co-deletion, contrary to *GCLc* (Fig 6A). Of note, none of these genes is present on the 1p/19q chromosome. Interestingly, when investigating the relationship between *CBS* and patient prognostic, a high *CBS* expression was associated with a better patient survival in IDHm-1p/19q co-deleted gliomas, suggesting that *CBS* can be used as a novel prognostic biomarker in this specific subgroup of gliomas (Fig 6B). Taken together, our data indicate that IDH mutation imposes a shortage of the reducing equivalents required to maintain a balanced redox state and may rely on cysteine availability for incorporation in GSH to ensure antioxidant functions. In this context, high *CBS* expression may represent a marker of oxidative stress and increased reliance on the transsulfuration pathway for cysteine generation. We suggest that this metabolic vulnerability may offer a therapeutic window for the treatment of IDHm glioma that awaits to be further exploited.

## Discussion

Despite considerable insight in the epigenetic alterations resulting from the gain-of-function mutation in IDH1 or IDH2, the metabolic consequences of the mutant enzyme, whether D2HD-dependent or not, are still poorly understood, in part due to the difficulties in establishing IDH-mutant gliomas in culture and/or *in vivo* as xenografts. Here, we applied *in situ* metabolic profiling and LC-MS on brain sections of glioma PDX and human glioma samples with and without the IDH1 mutation for large-scale unbiased metabolic profiling of these tumors. It should be noted that based on our current understanding, IDH-mutant and IDH wild-type gliomas represent biologically different entities and it is currently not clear which differences can be directly attributed to the IDH status. Key phenotypic differences, for example, proliferation rate and angiogenesis, are largely inherent to the tumor subtype and may also affect the metabolic signature. Notwithstanding this limitation, the IDH status is the primary determinant of glioma subtype classification (Louis *et al*, 2016) and interrogating the differences between IDHwt versus IDHm gliomas is essential to understand glioma biology. A key finding of our study is the major differences observed in phospholipid composition in IDHm gliomas, which is in line with recent observations made by MR spectroscopy (MRS) (Esmaeili *et al*, 2014; Jalbert *et al*, 2017). While MRS detects a signal from a pool of lipid molecules that resonate at the same frequency, while MSI can detect the distribution of specific metabolites that differ by their mass, which may explain differences obtained between the two approaches. Although the precise identification of the observed metabolites remains to be determined, it is interesting to speculate that such metabolites might be developed as markers of specific glioma subtypes using non-invasive imaging approaches. The exact mechanism through which mutant IDH affects lipid synthesis requires further insight, avenues lie in limited citrate availability for TCA cycle and fatty acid synthesis, and the inability of mutant IDH1 to engage in reductive carboxylation for citrate production (Grassian *et al*, 2014).

It is well known that IDHm gliomas are less aggressive than their wild-type counterparts. Here, we find a reduced energetic balance in

IDHm gliomas, which correlates with reduced turnover of glucose metabolism as shown by *in vivo* tracing of $^{13}C_6$-labeled glucose. Interestingly, recent data showed a direct inhibitory effect of D2HG on pyruvate dehydrogenase (PDH) (Izquierdo-Garcia *et al*, 2015) and ATP synthase activity (Fu *et al*, 2015). Reduced glycolysis is also in line with the reported silencing of lactate dehydrogenase A through promoter hypermethylation in IDH1m tumors (Chesnelong *et al*, 2014). Thus, the reduced energy potential may be the combined result of limited IDHwt function and of overproduction of D2HG by the mutant enzyme directly affecting metabolic enzyme activity and gene expression.

*In vivo* tracing showed minimal glucose-derived label in αKG and D2HG suggesting that glucose is not the prevalent carbon source for these metabolites in IDHm gliomas (Fig 3). This is in sharp contrast with the results obtained in IDHwt glioma and normal brain, where under the same experimental conditions both αKG and D2HG are, at least in part, derived from glucose. Although this might be explained by a high turnover of mitochondrial αKG, it raises the question of alternative substrates and/or compartmentalization effects. Acetate was recently shown to act as a bioenergetic substrate for glioblastoma (Mashimo *et al*, 2014), and NAA and NAAG supplemented to glioblastoma cells were found to support tumor growth (Long *et al*, 2013). NAA and NAAG are involved in the acetate metabolism which also regulates lipid synthesis and histone acetylation (Gao *et al*, 2016). In line with earlier data (Reitman *et al*, 2011), NAAG levels were heterogeneous in IDHm gliomas, suggesting differential activation of metabolic pathways. *In vivo* experiments with stable acetate tracers will be required to evaluate the contribution of different acetate sources as a bioenergetic fuel and source of D2HG in IDHm glioma.

In agreement with our previous work (Tardito *et al*, 2015), isotopically labeled glutamine intravenously injected in glioma xenografts was neither efficiently taken up by the brain nor by tumors independent of their IDH status (Fig EV3A); thus, it can be ruled out that blood-borne glutamine is a relevant source of carbons for D2HG production in IDHm glioma. This is in line with an earlier study indicating that blood-borne glutamine is not significantly metabolized in glioma xenografts (Marin-Valencia *et al*, 2012). Several *in vitro* studies suggested increased reliance of IDHm cells on glutamine-dependent anaplerosis (Reitman *et al*, 2011; Ohka *et al*, 2014). Transcriptional silencing or chemical inhibition of glutaminase, the enzyme converting glutamine to glutamate, led to reduced cell proliferation in IDHm cells *in vitro* (Seltzer *et al*, 2010; Emadi *et al*, 2014). However, this is not supported by our *in vivo* data since in both IDHwt and IDHm tumors, the cataplerotic activity of glutamine synthetase (GS), as opposed to the anaplerotic activity of glutaminase, was detectable ($^{13}C_2$ glutamine in Fig 3C). Moreover, in the clinical samples, the total level of glutamate was significantly reduced in IDHm gliomas, while glutamine was comparable in both tumor types and close to normal brain values. In the brain context, where glutamate can be provided by astrocytes, the importance of tumor-autonomous glutaminolysis appears to be minimal (Tardito *et al*, 2015) and the high glutamate concentrations in the brain may render gliomas less vulnerable to glucose-derived metabolic fluctuations (van Lith *et al*, 2014). Taken together these aspects highlight the importance of studying brain tumor metabolism *in vivo* within its organ-specific microenvironment that affects the metabolic behavior of glioma cells.

Earlier work on wild-type IDH demonstrated its crucial role in the maintenance of the cellular redox state, presumably through its NADPH-producing activity (Jo *et al*, 2001; Lee *et al*, 2002). In line with previous studies (Bleeker *et al*, 2010; Molenaar *et al*, 2015), we found a small reduction in NADPH activity in IDHm glioma *in vivo*, yet surprisingly GSH levels were maintained, which is in contrast to cell lines engineered to overexpress the mutant enzyme (Mohrenz *et al*, 2013; Shi *et al*, 2014, 2015). *De novo* GSH synthesis relies on cysteine availability, which can be replenished via extracellular import by the cystine-glutamate antiporter xCT or via intracellular synthesis through the transsulfuration pathway (McBean, 2012). Both routes are important for maintaining the antioxidant potential in cells under oxidative stress when the demand for GSH is elevated (Ogunrinu & Sontheimer, 2010; McBean, 2012). The increased expression of *CBS* and *GCLc* in IDHm gliomas supports the idea that IDHm tumors may preferentially utilize the transsulfuration pathway for GSH synthesis. The importance of CBS and GCL in tumorigenesis has recently been highlighted (Takano *et al*, 2014; Harris *et al*, 2015). Here, we identify *CBS* not only as a novel prognostic marker for 1p/19q co-deleted IDHm gliomas (ODG subtype), but also as a potential target to tilt the balance toward high-oxidative stress in IDHm gliomas. In conclusion, the present study provides important insight into the metabolism of IDHm and IDHwt gliomas and points to hitherto unrecognized metabolic vulnerabilities imposed by the activity of mutant IDH.

# Materials and Methods

### Patient samples

Glioma samples for PDX generation were collected at the Centre Hospitalier in Luxembourg (Neurosurgical Department, CHL) or the Haukeland University Hospital (Bergen, Norway) from patients who have given informed consent. Clinical samples for LC-MS analysis were obtained from 13 patients from the Haukeland Hospital, Bergen, Norway, where tissue fragments were stereotactically sampled (~25 mg) during the operation and snap-frozen. Diagnosis and IDH status of clinical samples were determined by classical neuropathology and immunohistochemistry for IDH1 (6 IDHm gliomas, 7 IDHwt gliomas). IDH1 status, 1p/19q co-deletion, hypermethylation status, and grade are indicated in Table EV6. D2HG production was verified by LC-MS confirming the IDHm status of the samples (Fig EV4). All studies were conducted according to the Declaration of Helsinki and with approval from the local ethics committees (National Ethics Committee for Research (CNER) Luxembourg (ADAPT protocol REC-LRNO-20110708) and local ethics committee Haukeland University Hospital, Bergen (REK 2010/130-2)).

### Patient-derived glioma xenografts

Patient-derived glioma xenografts (PDX) were generated in NOD/Scid mice (male or female, at least 2 months of age) as previously described (Golebiewska *et al*, 2013; Navis *et al*, 2013; Sanzey *et al*, 2015; Bougnaud *et al*, 2016) from 3 IDHwt gliomas (P3, P8, and T434) and 3 IDHm gliomas (E478, T186, and T394). Histopathological and molecular diagnostics are shown in Table 1. Orthotopic

xenografts were maintained by serial transplantation in the brain of mice, using either organotypic spheroids established in short-term culture (minced tissue fragments maintained for 7–10 days in flasks coated with 0.75% agar under standard tissue culture conditions) (Bougnaud *et al*, 2016) for IDHwt tumors, or by direct inoculation of fresh tumor material into the right frontal cortex (IDHm tumors). All mice were housed in an isolated specific-pathogen-free environment. Food and water were supplied *ad libitum*. Mice were sacrificed at the first signs of neurological symptoms (locomotor problems, uncontrolled movements) or behavioral abnormalities (prostration, hyperactivity); the brains were dissected out and cut into two pieces along the coronal plane at the tumor core. The half brains were immediately snap-frozen in isopentane/liquid nitrogen for preparation of either cryosections for MSI or for metabolite extraction for LC-MS. The handling of the animals and the surgical procedures were performed in accordance with the European Directive on animal experimentation (2010/63/EU) and were approved by the institutional and national authorities responsible for animal experiments in Luxembourg (Protocol LRNO-2014-03).

**Mass spectrometry imaging (MSI)**

Ten-micrometer-thick cryosections of glioma xenografts were prepared on a cryostat (Microme HM560), set at −21°C for specimen and blade parameters, and mounted on conductive slides (Indium Tin Oxide, ITO) for MSI and cryodesiccated for 30 min. Slides were then transferred into a desiccator under vacuum for 30 min to complete drying and scanned for next step (teaching with imaging software). Matrix deposit for MSI : 9-AA (9-Amino Acridine) matrix was prepared at 5 mg/ml in 100% MeOH (LC-MS grade) and sprayed onto the slide with the Suncollect sprayer (Sunchrom, GmbH Germany). Analyses were performed on a Matrix Assisted Laser Desorption Ionisation–Fourier Transform Ion Cyclotron Resonance instrument (MALDI-FTICR) 7.0 T (Bruker Daltonics GmbH, Bremen, Germany) in full scan negative mode within 70–1,000 Da mass range at 100-μm spatial resolution. The horizontal axis in a mass spectrum is expressed in units of *m/z*, which represents the mass over charge (number of ions) ratio. For the untargeted analysis, 50 pixels of analysis from the imaging data (i.e., 50 mass spectra) per region of interest (ROI) were selected to run a statistical test on ClinProTools 3.0 software (Bruker, Germany) (Table EV1) using a threshold of signal to noise of 5 to compare the peaks between conditions. Recalibration, average peak list calculation, and peak calculation from ClinProTools allowed obtaining statistical results with metabolites using *t*-test. Relevant metabolites were considered for *P* value < 0.01 (from *t*-test) and fold change > 3 between conditions (Tables EV2 and EV3). MSI data were operator-verified to check the imaging data and suppress false positives. Imabiotech metabolite database as well as Metlin and the Human Metabolome DataBase (HMDB) were interrogated for identification of the compounds found with ClinProTools. Mass accuracy was set at 10 ppm, and lists of metabolites were established with their structures. When several hits matched with the *m/z* of interest, one representative compound was selected for presentation in Tables EV2 and EV3. For the targeted approach, measurement regions included identical large ROIs selected in biological sections and were analyzed by Fleximaging 4.0 software (Bruker Daltonics GmbH, Germany). Initially for each condition (PDX or control brain

without implantation), three sections from three animals were analyzed for quantification. In a second step, sections were analyzed from six different PDX models. To ensure the correct assignment of specific metabolites, we performed fragmentation analysis to confirm their identity and were able to validate nine of 16 compounds including glutamate (Glu), glutamine (Gln), N-acetyl aspartate (NAA), N-acetyl aspartyl glutamate (NAAG), cystathionine, reduced glutathione (GSH), ascorbic acid, citric acid, and cytidine (Table EV4). Metabolites were validated if at least one fragment was found to be relevant.

LESA-nESI-FTICR was applied for specific metabolites that were undetectable by MALDI as described previously (Navis *et al*, 2013). A Nanomate Triversa (Advion) in "Liquid Extraction Surface Analysis" (LESA) mode coupled to a nano-ElectroSpray Ionisation (nano-ESI) source on the FTICR was used to extract analytes from biological sections with an extraction solution. Sample plates were cooled down to 12°C during analyses to limit degradation or secondary reactions. Fragmentation was performed in positive and negative mode within the collision cell of the SolariX after isolation of the parent compound in the quadrupole to confirm the identity of the metabolites. Spray parameters were set as follows: voltage to apply 1.30 kV and gas pressure 0.40 psi. Extraction solvent consisted of 65:15:20 MeOH:IPA: water + 5 mM ammonium acetate using LC-MS quality solvents. Isolation of the ion in the quadrupole within a 2–5 mDa window and fragmentation with appropriate collision energies in the collision cell (qCID) (±15 V). After MALDI analysis, the 9-AA matrix was removed from the slide with MeOH and hematoxylin–eosin (H&E) staining was performed to visualize the tumor area. This image was later implemented into Quantinetix™ software for data analysis by overlaying the optical image with molecular images. An intensity-dependent color code shows the relative amount of a specific compound (*m/z* value) throughout the tissue section, which was also used for quantification (log intensity values).

**Isotopically labeled reference compounds and tissue extinction coefficient for MSI**

The signal obtained by MALDI on tissue sections can be modulated by the chemical composition of the tissue and thus can differ from one tissue to another. This can be controlled for by determining the tissue extinction coefficient (TEC) for each metabolite. Since the metabolites are endogenous molecules in the brain, labeled forms had to be used in this approach. Therefore, for the TEC determination, isotopically labeled compounds were mixed with the 9-AA matrix (5 mg/ml in MeOH) to achieve 20 μM final concentration, and the final mixture was sprayed on the entire slide for analysis. For TEC analysis, an additional region was selected outside tissue sections as a control area (no tissue effect in this area) for calculations of TEC values, defined as the ratio of compound signal in the tissue section to the compound signal outside the tissue section. The following compounds were analyzed: NAA-d3, NAAG-d3 (Spaglumic acid-d3), L-cystathionine-d4, GSH $^{13}C$, $^{15}N$ and GSSG $^{13}C$,$^{15}N$. NAA-d3 was purchased from CDN isotopes, all other itotopically labeled compounds from Toronto Research Chemical, Canada. LC-MS grade solvents were used for sample preparation. Overall, the TEC values were comparable for the different regions (mutant or wild-type tumor, contralateral brain, and control brain), indicating

that for the majority of compounds, the quantification is not likely to be affected by strong tissue effects (Table EV5). Except for GSSG, which was excluded from the present data, since all isotopic compounds were simultaneously present in the mix, oxidation reactions cannot be excluded, which may explain differences in GSSG values. Data were treated with Fleximaging 4.0 (*Bruker, Germany*), proprietary softwares Quantinetix™ 1.7, and Multimaging™ 1.0.30 (*ImaBiotech, Loos, France*).

### LC-MS analysis

LC-MS analysis was run on snap-frozen tissue of six PDX samples and 13 clinical samples. Each sample was analyzed in three replicates. For metabolite extraction, a 5-mm metal bead (Qiagen, No69989) and extraction buffer (methanol/acetonitrile/water-50/30/20) containing 100 ng/ml HEPES for internal standard purpose (Sigma, H4034) were added to the tissue at a volume/weight ratio of 25 μl/mg. The tissue was disrupted in a bead mill (Qiagen, Tissuelyzer, Hombrechtikan, Switzerland) with 2 cycles (2 × 20 s/20 MHz) before vortexing on an Eppendorf Thermomixer at 2°C/20 min/1,400 rpm. The extract was clarified by centrifugation (15 min/13,362 *g*/4°C in an Eppendorf centrifuge) and transferred in fresh tubes and stored at −80°C. LC-MS analysis was performed as previously described (Tardito *et al*, 2015). Tumor extracts were compared to extracts from contralateral mouse brain for reliable comparison between the two models.

### Statistical analysis of MS data

For statistical analysis of mass spectrometry (MS) data, we used the standard tests implemented in the "stats" package of R. In order to be able to apply parametric statistical tests, the MS intensity data were log-transformed. To test the applicability of the normality assumption, we first normalized log intensities between metabolites (by mean centering and scaling) and then submitted the united data vector to Shapiro and Kolmogorov–Smirnov tests. Both parametrical tests (ANOVA and unpaired *t*-test) and non-parametrical Wilcoxon (Mann–Whitney) test were used to compare mean values of log intensities for MSI and LC-MS data depending on data quantity and number of compared groups. Tukey's honest significant differences were used for post hoc analysis of ANOVA results for LC-MS data (not shown). In order to compare the large sets of MSI data between several samples, we applied Wilcoxon test with the most stringent Bonferroni adjustment. As, despite this stringency, the large number of data points in MSI data may lead to overestimation of the significance, mean MSI intensities were considered as well. For all MS data shown, *P*-values are based on unpaired two-tailed *t*-test of mean abundances of metabolites between PDX with and without IDH1 mutation. Whenever error bars are shown in the figures with metabolite abundances, they correspond to standard errors of the mean, except for box plots that follow the classical visualization rules and are based on quartiles.

Testing normality of the MSI data gave the *P*-value of 0.46 and 0.98 for Shapiro and Kolmogorov–Smirnov tests concordantly, suggesting that the log-transformed data may come from the normal distribution. For global LC-MS data, we observed low *P*-value for Shapiro test (*P*-value = 2.5e-7) and non-significant Kolmogorov–Smirnov test (*P*-value = 0.24). However, violations of

**The paper explained**

**Problem**

Gliomas are malignant brain tumors which are currently incurable. Point mutations in the enzyme isocitrate dehydrogenase (IDH) are thought to be a driver for a major subset of gliomas, leading to the generation of the oncometabolite D2HG and subsequent abnormalities in gene expression. The metabolic consequences of the mutation are less well understood.

**Results**

The key results of the present work include the following: (i) novel insight into aberrant lipid metabolism in IDH-mutant gliomas, (ii) demonstration of low energy potential in IDH-mutant gliomas, (iii) identification of potential compensatory pathways to maintain redox balance and establishing a novel prognostic factor in patients with oligodendroglial glioma subtype.

**Impact**

The present study uncovers a mutant IDH-specific metabolic landscape which is clinically relevant and highlights novel metabolic vulnerabilities in IDH-mutant gliomas that may be therapeutically exploited.

normality were caused by a single sample (P3_NC1331): If it is excluded, both tests do not show significant deviation from normality (Shapiro's *P*-value = 0.062, Kolmogorov–Smirnov's *P*-value = 0.083).

### Administration of $^{13}$C-labeled tracers

U-$^{13}$C$_6$-Glucose ($^{13}$C$_6$-Glc) or U-$^{13}$C$_5$ glutamine ($^{13}$C$_5$-Gln) tracer from Cambridge Isotope Laboratories (CIL, USA, Andover, MA) was dissolved in saline solution (0.9% NaCl) and administered by a single bolus injection in the tail vein to ensure doses of 1 mg/g for the $^{13}$C$_6$-Glc or 0.15 mg/g for $^{13}$C$_5$-Gln. The animals were sacrificed 22 min after tracer administration, and brains were cryopreserved in isopentane for LC-MS analysis. For each PDX, three xenografted mice were analyzed per tracer.

### NADP/NADPH quantification

Relative quantification of the oxidized and reduced adenine dinucleotide phosphates (NADP$^+$ and NADPH) in tissue extracts was obtained with the bioluminescent NADP/NADPH-Glo™ Assay (Promega, G9081). For comparative analysis, we used a standardized extraction procedure with a bead mill (TissueLyser II, Qiagen, 20 Hz/20 s) for tissue disruption and metabolite extraction and a buffer/tissue ratio of 50 μl of the recommended extraction buffer per milligram of tissue. Luminescence signals were measured in a 96-well format in a Lumistar plate reader. Total oxidized and reduced nicotinamide amounts expressed in relative luminescence units served to determine NADP/NADPH ratios in tissue extracts. Calibration of the assay was done with a NADP$^+$ standard curve.

### TCGA analysis

Glioma patient data of the TCGA cohort (fused LGG and GBM (Ceccarelli *et al*, 2016)) were obtained from cBioPortal (Gao *et al*,

2013) and analyzed using R software. Analysis of TCGA database cohort was performed as follows: Expression data were based on RNA-Seq for IDHwt ($n = 220$), for IDHm with 1p/19q co-deletion ($n = 151$), and IDHm without co-deletion ($n = 226$), total $n = 597$. Center lines of the box plots show the medians; box limits indicate the 25th and 75th percentiles; outliers are represented by dots. Notches indicate $\pm 1.58 \times$ IQR/sqrt(n) and give 95% confidence that two medians differ (Chambers *et al*, 1983). For Kaplan–Meier survival analyses, gene expression median was calculated independently for each clinical group. Comparison of survival curves was done using log-rank (Mantel–Cox) test, using GraphPad Prism software.

**Expanded View** for this article is available online.

## Acknowledgements

The authors thank Virginie Baus for excellent technical assistance with the animal work and acknowledge the financial contribution of the Luxembourg Institute of Health (Luxembourg) and the Stiftelsen Kristian Gerhard Jebsen (Norway). The authors would like to dedicate this publication to our dear colleague Chantal Courtois, who passed away during the resubmission of this work.

## Author contributions

SPN, RB, JS designed the research studies; GH, ST, FF, AO, AB, LZ conducted experiments; GH, ST, LZ, and ACH acquired data; FF, GH, ST, OK, SF, A-CH, SPN analyzed data; PVN performed statistics, FF, ST, EG, AG, SPN, WL interpreted data; WL provided material; RB and ML-J established the clinical sample collection; SPN, ST, FF, AG wrote and prepared the manuscript. SPN supervised the study. All authors revised and approved the manuscript.

## Conflict of interest

GH and JS are employed by Imabiotech. All other authors report no conflict of interest.

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
