## [Review Process File · EMBO Molecular Medicine]

Altered metabolic landscape in IDH mutant gliomas affects phospholipid, energy and oxidative stress pathways

Fred Fack, Saverio Tardito, Guillaume Hochart, Anais Oudin, Liang Zheng, Sabrina Fritah, Anna Golebiewska, Petr V. Nazarov, Amandine Bernard, Ann-Christin Hau, Olivier Keunen, William Leenders, Morten Lund-Johansen, Jonathan Stauber, Eyal Gottlieb, Rolf Bjerkvig and Simone P. Niclou

Corresponding author: Simone Niclou, Luxembourg Institute of Health

Review timeline:

Submission date:	06 March 2017
Editorial Decision:	11 April 2017
Revision received:	24 July 2017
Editorial Decision:	29 August 2017
Revision received:	10 September 2017
Accepted:	12 September 2017

Transaction Report:

Editor: Roberto Buccione

1st Editorial Decision

11 April 2017

Thank you for the submission of your manuscript to EMBO Molecular Medicine. We have now heard back from the Reviewers whom we asked to evaluate your manuscript.

I again apologise for the delay in reaching a decision on your manuscript. In this case, we first experienced difficulties in securing expert and willing reviewers. Further to this I was presented with a difficult decision, which required extended discussion and consultation with an additional external advisor who was not immediately available.

As you will see, all three reviewers are of the opinion that this is a very interesting study that addresses an extremely important issue, namely the functional consequences of IDH mutations in gliomas, with state of the art approaches and analysis. Reviewer 1 and 2 are more positive, although a number of important concerns are expressed, especially by reviewer 2, including the lack of acceptable statistical analysis and details on experimentation, issues in interpretation of the correlation between survival and gene expression levels, and other. Reviewer 3 however finds that, in addition to some of the concerns also expressed by reviewer 2, the study has an insurmountable flaw that compromises its value and eligibility for publication. Specifically s/he notes that the study compares oligodendrogliomas as mutant IDH tumors to glioblastomas as wt IDH controls, and thus compares different tumor types that differ in multiple parameters in addition to the IDH status.

Our reviewer cross-commenting exercise emphasized the latter issue, with reviewer 3 confirming his/her stance, and reviewer 1 actually converging on the fact that the issue is indeed a serious one

that must be addressed.

As I mentioned above, after further internal discussion, I sought further advice from an additional expert, who was not immediately available. S/he agreed that the study is interesting, addresses an extremely important issue and is based on advanced methodology. However, the expert ultimately also agreed that "IDHwt glioblastoma cannot be considered as an adequate control for IDHmt oligodendroglioma since these are completely different entities with a plethora of differences in patient age, tumor location, survival, genetic alterations, disturbances in tumor microenvironment etc. As such, one would expect huge metabolic differences in these two tumor entities that could be assigned to any of the above-mentioned parameters, in particular genetic alterations and differences in the tumor microenvironment... it is not possible, in my view, to account for the observed differences in metabolomics to mutations in IDH alone, as the authors pursue". The advisor also admitted that it is indeed a challenge to choose the most appropriate control for this study. For instance, "Since oligodendrogliomas are actually defined as IDHmt/1p19q co-deleted, IDHwt oligodendrogliomas (in case they exist) are not an adequate control either. Probably the best control is, as discussed by reviewer 1, to engineer an IDH1mt oligodendroglioma to IDHwt and directly compare these two tumors"

In conclusion, while publication of the paper cannot be considered at this stage, given the potential interest of your findings and after internal discussion, we have decided to give you the opportunity to address the criticisms.

We are thus prepared to consider a substantially revised submission, with the understanding that the Reviewers' concerns must be addressed with additional experimental data where appropriate, especially concerning the lack of an adequate control for IDHmt oligodendroglioma, and that acceptance of the manuscript will entail a second round of review. The overall aim is to significantly upgrade the relevance and conclusiveness of the dataset, which of course is of paramount importance for our title.

Please note that it is EMBO Molecular Medicine policy to allow a single round of revision only and that, therefore, acceptance or rejection of the manuscript will depend on the completeness of your responses included in the next, final version of the manuscript.

Since as mentioned above, the required revision in this case appears to require a significant amount of time, additional work and experimentation, and might be technically challenging, I would understand if you chose to rather seek publication elsewhere at this stage. Should you do so, we would welcome a message to this effect.

EMBO Molecular Medicine now requires a complete author checklist (<http://embomolmed.embopress.org/authorguide#editorial3>) to be submitted with all revised manuscripts. Provision of the author checklist is mandatory at revision stage; the checklist is designed to enhance and standardize reporting of key information in research papers and to support reanalysis and repetition of experiments by the community. The list covers key information for figure panels and captions and focuses on statistics, the reporting of reagents, animal models and human subject-derived data, as well as guidance to optimise data accessibility. This checklist especially relevant in this case given the issues raised with respect to statistical treatment and animal numbers.

As you know, EMBO Molecular Medicine has a "scooping protection" policy, whereby similar findings that are published by others during review or revision are not a criterion for rejection. However, I do ask you to get in touch with us after three months if you have not completed your revision, to update us on the status. Please also contact us as soon as possible if similar work is published elsewhere.

***** Reviewer's comments *****

Referee #1 (Comments on Novelty/Model System):

Patient-derived oligodendroglioma xenograft models with 1p/19Q deletions and defined IDH1

mutation status. Outstanding resources.

Referee #1 (Remarks):

Summary. Analytical chemistry methods (mass spectrometry imaging, and in vivo tracing of labeled nutrients by LC-MS) are applied to profile metabolic states in oligodendroglial tumors (tissue specimens and PDX models) contrasted by the involvement of wild type versus mutant IDH1. The phospholipid composition differs markedly dependent on IDH mutation status. Tumors with mtIDH1 have a low energy potential and high oxidative state, implying vulnerability to redox imbalance. Alternative carbon sources appear to be involved in mtIDH1 tumors producing the oncometabolite, D2HG. Genes contributing to the glutathione synthesis pathway are overexpressed (highly) in mtIDH1 tumors; expression of cystationine-beta-synthetase, specifically, correlates with patient survival.

Critique. Fack et al assemble high-quality metabolic portraits of oligodendroglial tumors segregated by the presence or absence of mutant IDH1. The significance of the report lies in the well-appreciated correlation between mtIDH1/2 and GCIPM signature, as well as the largely unexplored link between oncometabolites, glutathione synthesis pathway, and the natural history and molecular pathology of this class of tumors. The findings illustrate remarkable segregation of wild type from mutant IDH1 glial tumors.

The manuscript Introduces the state of the field with current knowledge as well as with probing unknowns, setting a stage for the direct, but elaborate, investigation. Relevant preclinical models of oligodendroglioma (some unique to this research team) are deployed with candid and precise explanations.

Six figures, six tables (supplemental), and four supplemental figures accrue an outstanding resource for setting the bar of oncometabolite biology research, and assessing consequences to tumor progression. MALDI (Figs 1 and 2) set the stage for obvious metabolic contrasts of the tumor types, while Fig 3 portrays the effects to the TCA Cycle, the glutamate pathway (Fig 4), and redox biology (Fig 5). Figure 6 brings out the impact on patient survival attributable to metabolic differences in mtIDH1 versus wtIDH1 oligodendrogliomas.

The Discussion places the new findings into a broader context of neuro-Oncology and glioma biology, with provocative leads for treatment opportunities.

Referee #2 (Remarks):

Fack and colleagues aimed to understand the impact of IDH mutation on the metabolism of glioma samples. They used mass spectrometry imaging and liquid chromatography/mass spectrometry to profile the levels of more than 100 metabolites in an IDH mutant glioma samples, an IDH wild type glioblastoma sample, and a normal brain tissue. The resulting dataset rendered evidence that supports differences in energy potential and oxidative phosphorylation levels between the samples. The study provides much needed insights into what may present potential metabolic vulnerabilities. The main weakness is the lack of statistical testing and associated significance in some of the findings due to small sample size.

Major comments

1. Metabolite profiling results are based on a single (?) IDH mutant oligodendroglioma derived neurosphere and it is not clear how representative these are for the general patient population of this glioma subtype. How can we be sure that differences observed are not due to technical or intratumoral biases?
2. Could individual results that seem of strong interest, such as the overrepresentation of phosphatidylethanolamine, be quantified in additional relevant samples?

3. It is not described what m/z ratios represent. The only place where this value is explain is in the legend (!) of the Supplementary Figure 1.
4. Comparisons between metabolite levels in clinical samples, i.e. glutamine, glutamate, should be tested for statistical significance.
5. Were clinical samples confirmed to be of the same glioma subtype as the parental tumors used in the xenografts, i.e. IDH wild type respectively IDH mutant with 1p/19q co-deletion?
6. The diagrams shown in Figure 3B, 5D could be enhanced using color codes to indicate the levels measured in the different experiments. At current they are uninformative, other than to show the text book version of the pathways.
7. Could the results obtained here be used to identify markers of IDH mutant vs IDH wild type glioma that could be measured using non-invasive imaging approaches?
8. Correlation between survival and gene expression levels is not an acceptable approach to determine outcome associations as it does not consider censoring. This analysis should involve a log-rank statistic derived from a Cox proportional hazards model, possibly in combination with a Kaplan-Meier curve for visualization of the results. The patient sets should at least be separated for codel status.
9. The Discussion is too long, and should be condensed.

Referee #3 (Comments on Novelty/Model System):

The study has a fatal flaw: it compares oligodendrogliomas to primary glioblastoma and attributes all metabolic differences to differences in the IDH mutation status, when these tumors have dozens of others genetic, molecular and other differences that can account for differences in metabolism. To be interpretable, the entire study would have to be repeated with a different and adequate control.

Referee #3 (Remarks):

The study by Fack et al. investigated the metabolic differences between IDH mutant (mIDH) and IDH wildtype (wtIDH) gliomas. The study used mass spectrometry imaging and in vivo tracing of labeled nutrients followed by liquid chromatography-mass spectrometry to compare the metabolic profiles of PDX tumors derived from mIDH oligodendrogliomas and wtIDH glioblastomas. It found differences in phospholipid composition, energy potential and oxidative state, enzymes required for de novo glutathione synthesis and other metabolites. Attempts at validation of select findings in clinical specimens were also conducted.

The stated goal of investigating metabolic changes induced by IDH mutations in gliomas is novel and significant in view of the incidence and not well understood role of IDH mutations in glioma genesis and malignancy. However, this goal is not achieved in the present study due to the following major weaknesses in study design and data analysis and interpretation:

1- The study compares oligodendrogliomas as mIDH tumors to glioblastomas as wtIDH controls. It is therefore comparing oligodendroglioma to astrocytoma, grade II/III gliomas to grade IV glioma, and secondary to primary tumors. In other words, it is comparing two different tumor types that differ in dozens (if not hundreds) of parameters (genetic and other molecular alterations, potential cell of origin, pathobiology and other) in addition to the IDH status. IDH status is not manipulated by any means. Yet, the study attributes all metabolic changes to the IDH status. This is a major flaw in the design of the entire study that leads to a complete mis/over-interpretation of all presented data.

2- Even ignoring the above major weakness, some of the presented data are over interpreted, in large part due to the general absence of statistical analyses. For example, the study describes an overall reduction of metabolites (lactate, malate, aspartate,...) incorporating heavy glucose carbons in mIDH vs wtIDH tumors (Fig. 3C). This reduction is by no means apparent in the figure (the graphs

look of almost identical magnitudes) and no numbers or statistics are given.

3- Contrary to what is stated, the limited human and TCGA data mostly do not validate the PDX-derived data as they sometimes show opposite results (e.g. glutamate and NAAG in Fig. B/C) or measure other parameters that are only indirectly associated with what was measured in the PDX (e.g. assessing the expression of SLC7A1, GCLC and CBS in TCGA but not in the PDX).

4- The study is entirely descriptive and correlative. It lacks functional and mechanistic data and has no clear message.

1st Revision - authors' response

24 July 2017

RESPONSE TO REVIEWERS' COMMENTS

Referee #1 (Comments on Novelty/Model System):

Patient-derived oligodendroglioma xenograft models with 1p/19Q deletions and defined IDH1 mutation status. Outstanding resources.

Response: We appreciate the recognition by Referee 1 of the unique animal models of patient-derived IDH mutant gliomas described here. Indeed these PDX models are extremely rare and only a handful of labs have succeeded in establishing and maintaining them. They represent crucial experimental tools to study glioma biology and therefore we believe the data presented here and the models should be made available to the scientific community. We have now expanded our set of IDH mutant PDX models to three and these have been integrated in the revised manuscript.

Referee #1 (Remarks):

Summary. Analytical chemistry methods (mass spectrometry imaging, and in vivo tracing of labeled nutrients by LC-MS) are applied to profile metabolic states in oligodendroglial tumors (tissue specimens and PDX models) contrasted by the involvement of wild type versus mutant IDH1. The phospholipid composition differs markedly dependent on IDH mutation status. Tumors with mtIDH1 have a low energy potential and high oxidative state, implying vulnerability to redox imbalance. Alternative carbon sources appear to be involved in mtIDH1 tumors producing the oncometabolite, D2HG. Genes contributing to the glutathione synthesis pathway are overexpressed (highly) in mtIDH1 tumors; expression of cystationine-beta-synthase, specifically, correlates with patient survival.

Critique. Fack et al assemble high-quality metabolic portraits of oligodendroglial tumors segregated by the presence or absence of mutant IDH1. The significance of the report lies in the well-appreciated correlation between mtIDH1/2 and GCIPM signature, as well as the largely unexplored link between oncometabolites, glutathione synthesis pathway, and the natural history and molecular pathology of this class of tumors. The findings illustrate remarkable segregation of wild type from mutant IDH1 glial tumors.

The manuscript Introduces the state of the field with current knowledge as well as with probing unknowns, setting a stage for the direct, but elaborate, investigation. Relevant preclinical models of oligodendroglioma (some unique to this research team) are deployed with candid and precise explanations.

Six figures, six tables (supplemental), and four supplemental figures accrue an outstanding resource for setting the bar of oncometabolite biology research, and assessing consequences to tumor progression. MALDI (Figs 1 and 2) set the stage for obvious metabolic contrasts of the tumor types, while Fig 3 portrays the effects to the TCA Cycle, the glutamate pathway (Fig 4), and redox biology (Fig 5). Figure 6 brings out the impact on patient survival attributable to metabolic differences in mtIDH1 versus wtIDH1 oligodendroglomas.

The Discussion places the new findings into a broader context of neuro-Oncology and glioma biology, with provocative leads for treatment opportunities.

Response: We thank Referee 1 for the positive evaluation of our work and the appreciation of the innovative approaches applied.

Referee #2 (Remarks):

Fack and colleagues aimed to understand the impact of IDH mutation on the metabolism of glioma samples. They used mass spectrometry imaging and liquid chromatography/mass spectrometry to profile the levels of more than 100 metabolites in an IDH mutant glioma samples, an IDH wild type glioblastoma sample, and a normal brain tissue. The resulting dataset rendered evidence that supports differences in energy potential and oxidative phosphorylation levels between the samples. The study provides much needed insights into what may present potential metabolic vulnerabilities. The main weakness is the lack of statistical testing and associated significance in some of the findings due to small sample size.

Major comments

1. Metabolite profiling results are based on a single (?) IDH mutant oligodendroglioma derived neurosphere and it is not clear how representative these are for the general patient population of this glioma subtype. How can we be sure that differences observed are not due to technical or intratumoral biases?

Response: We have now redone the analysis in six patient-derived glioma models, of which three are IDH mutant and three are IDH wildtype. All analyses were repeated both by MSI and by LC-MS and appropriate statistical analysis has been incorporated throughout.

2. Could individual results that seem of strong interest, such as the overrepresentation of phosphatidylethanolamine, be quantified in additional relevant samples?

Response: As indicated above, this has now been expanded to additional samples (6 in total) (see Fig. 1)

3. It is not described what m/z ratios represent. The only place where this value is explain is in the legend (!) of the Supplementary Figure 1.

Response: We apologize for this omission, the m/z ratio has now been clearly defined in the M&M section (page 12-13), as well as in the legends of the main figures.

4. Comparisons between metabolite levels in clinical samples, i.e. glutamine, glutamate, should be tested for statistical significance.

Response: Statistical analysis has now been included for all analyses and a paragraph on statistical methods has been included in the M&M section (page 14).

5. Were clinical samples confirmed to be of the same glioma subtype as the parental tumors used in the xenografts, i.e. IDH wild type respectively IDH mutant with 1p/19q co-deletion?

Response: The diagnostic subtype of all clinical samples is shown in Suppl. Table 6. The clinical samples include the same glioma subtypes as shown for xenografts, namely IDH mutant (ODGs, astrocytoma and GBM) and IDH wildtype (GBMs only).

6. The diagrams shown in Figure 3B, 5D could be enhanced using color codes to indicate the levels measured in the different experiments. At current they are uninformative, other than to show the text book version of the pathways.

Response: The schematic (Fig. 3B) is meant to highlight the expected labels that are shown in Fig. 3C, we believe that this is helpful for the reader to be able to follow the origin of the labels in Fig. 3C. For a better visualization, we have now added a separate quantification of the intensities of C₂ and C₃ labeled compounds as Fig. 3D.

7. Could the results obtained here be used to identify markers of IDH mutant vs IDH wild type glioma that could be measured using non-invasive imaging approaches?

Response: We thank the reviewer for this interesting remark. This is surely a possibility since some of the metabolites might be detectable by MR spectroscopy and thus would allow for translation as non-invasive biomarker into the clinical setting. This has now been added to the discussion (page 9). It should be noted however, that this would need substantial validation in the MRS setting, particularly since MRS is less sensitive than the approaches used here and a direct translation may prove challenging.

8. Correlation between survival and gene expression levels is not an acceptable approach to determine outcome associations as it does not consider censoring. This analysis should involve a log-rank statistic derived from a Cox proportional hazards model, possibly in combination with a Kaplan-Meier curve for visualization of the results. The patient sets should at least be separated for codel status.

Response: Our analysis initially derived from “the regulome explorer web tool”, which does not consider censoring, but bases correlation plots on continuous expression of a gene rather than on median expression clinical subgroup definition. Also, we initially did not provide Kaplan Meier data for CBS expression in all gliomas, as it would rather reflect the positive effect of IDH mutation on patient prognosis. In the revised manuscript we now apply the more standard approach of separating patients in low and high expressing cohorts and used Log-rank (Mantel-Cox model) test and Kaplan-Meier curve for visualization. We further separated the patients according to the 1p/19q chromosomal codel status for gene expression and for survival analyses. We thank the reviewer for this valuable comment as it provided novel insight for the 1p19q co-deleted patient subgroup (Fig. 6).

9. The Discussion is too long, and should be condensed.

Response: We thank the reviewer for pointing this out and have now condensed the discussion by almost one third (from originally 1667 words to 1164 words).

Referee #3 (Comments on Novelty/Model System):

The study has a fatal flaw: it compares oligodendrogliomas to primary glioblastoma and attributes all metabolic differences to differences in the IDH mutation status, when these tumors have dozens of others genetic, molecular and other differences that can account for differences in metabolism. To be interpretable, the entire study would have to be repeated with a different and adequate control.

Response: We understand the reviewer’s concern with regard to the appropriate control for IDH mutant gliomas. However we cannot accept this as ‘a fatal flaw’ since the vast majority of published data and studies involving glioma patients are based on comparisons between these different glioma entities. It should be noted that gliomas have only recently been re-classified with the IDH status being the primary determinant of glioma subtyping. Thus IDH status defines specific glioma subtypes and the aim of the present manuscript was to compare the metabolic differences inherent to these subtypes. In order to address the issue of appropriate control, we have now increased the number of samples per subtype and added a relatively rarely occurring GBM carrying the IDH mutation. The latter should thus allow to distinguish the effect of grade versus IDH status and this has been highlighted in the revised manuscript where appropriate.

Finally we would like to emphasize that we do not pursue that all observed differences in metabolomics are solely due to IDH status. This is obviously impossible to conclude based on the multiple aberrations seen in these tumors, as correctly pointed out by the reviewer. Nevertheless, being an initial driver mutation occurring early in the tumorigenic process, it can be assumed that many abnormalities resulted as direct or indirect consequences of the IDH mutation, as any subsequent event may be related to the initial mutation. To elucidate this was however not the purpose of our study as we only aimed to address the final outcome (differences) of IDH mutant versus IDH wildtype gliomas as observed in the patient. Eventually it is these differences that are

important for therapeutic interventions, independent of the direct culprit that has led to these differences. We have clarified this issue in the revised manuscript.

Referee #3 (Remarks):

The study by Fack et al. investigated the metabolic differences between IDH mutant (mIDH) and IDH wildtype (wtIDH) gliomas. The study used mass spectrometry imaging and in vivo tracing of labeled nutrients followed by liquid chromatography-mass spectrometry to compare the metabolic profiles of PDX tumors derived from mIDH oligodendrogliomas and wtIDH glioblastomas. It found differences in phospholipid composition, energy potential and oxidative state, enzymes required for de novo glutathione synthesis and other metabolites. Attempts at validation of select findings in clinical specimens were also conducted.

The stated goal of investigating metabolic changes induced by IDH mutations in gliomas is novel and significant in view of the incidence and not well understood role of IDH mutations in glioma genesis and malignancy. However, this goal is not achieved in the present study due to the following major weaknesses in study design and data analysis and interpretation:

1- The study compares oligodendrogliomas as mIDH tumors to glioblastomas as wtIDH controls. It is therefore comparing oligodendroglioma to astrocytoma, grade II/III gliomas to grade IV glioma, and secondary to primary tumors. In other words, it is comparing two different tumor types that differ in dozens (if not hundreds) of parameters (genetic and other molecular alterations, potential cell of origin, pathobiology and other) in addition to the IDH status. IDH status is not manipulated by any means. Yet, the study attributes all metabolic changes to the IDH status. This is a major flaw in the design of the entire study that leads to a complete mis/over-interpretation of all presented data.

Response: Please see above. We would also like to emphasize that this study is conducted on clinical samples only (patient-derived xenografts and biopsies). There is no culture model available to study these tumors and as such it is technically not possible to undertake genetic manipulations in these samples. The strength of our study is precisely that it relies on human glioma samples with an endogenous IDH mutation in the expected genetic background, and not on e.g. GBM cell lines where an IDH mutation was introduced.

Please kindly note that we do nowhere pretend that all observed differences are merely attributable to the IDH mutant enzyme, in fact this has been clearly highlighted on several occasions in the manuscript (e.g. Discussion section page 11).

2- Even ignoring the above major weakness, some of the presented data are over interpreted, in large part due to the general absence of statistical analyses. For example, the study describes an overall reduction of metabolites (lactate, malate, aspartate,...) incorporating heavy glucose carbons in mIDH vs wtIDH tumors (Fig. 3C). This reduction is by no means apparent in the figure (the graphs look of almost identical magnitudes) and no numbers or statistics are given.

Response: We thank the reviewer for these valuable remarks. We have incorporated appropriate statistical analysis for all samples in the revised manuscript. All data analysis was redone on 6 samples for MSI and LC-MS (with three technical replicates per sample in the LC-MS analysis) and the results with statistical analyses are now clearly reported in all figures (and suppl. figures). We have applied very stringent statistical analysis (see M&M section page 14) and have further been very cautious not to over-interpret the findings.

3- Contrary to what is stated, the limited human and TCGA data mostly do not validate the PDX-derived data as they sometimes show opposite results (e.g. glutamate and NAAG in Fig. B/C) or measure other parameters that are only indirectly associated with what was measured in the PDX (e.g. assessing the expression of SLC7A1, GCLC and CBS in TCGA but not in the PDX).

Response: We apologize if there was some confusion in the presentation of the clinical data. In fact we sometimes do see considerable variations between individual patient data (e.g. NAAG), which is seen both in PDX and in patients (and is also highlighted in the text). Unfortunately we could not analyse all metabolites in the patient material, since here we could not apply MALDI imaging, the latter requiring fresh-frozen sections. Nevertheless the majority of the presented key metabolites

showed a similar profile in patient samples and if this was not the case we have clearly indicated this in the text (e.g. glutamate, page 7). For clarification, all available clinical data is now shown in one figure (Fig. EV4).

4- The study is entirely descriptive and correlative. It lacks functional and mechanistic data and has no clear message.

Response: The description of an unknown biological phenomena is both novel and informative and is part of all scientific discoveries. We would like to stress here the novelty of our data, both in terms of biological samples and in terms of approaches. It should be noted that there are no patient-derived IDH mutant animal models (let alone cellular models) available in the scientific community and thus very little is known about the metabolic behaviour of these tumors. The present work provides key information to understand the importance of specific pathways involved in IDH mutant glioma biology and paves the way for important downstream functional studies.

2nd Editorial Decision

29 August 2017

Thank you for the submission of your revised manuscript to EMBO Molecular Medicine. We have now received the enclosed reports from the reviewers that were asked to re-assess it. As you will see the reviewers are now supportive, although reviewer #2 has a few final suggestions, with which we agree, that require your action. Please note that reviewer 2's point 4 is addressed in my editorial requirements list further below.

I am thus prepared to accept your manuscript for publication pending satisfactory compliance with the reviewer's final requests. Please also fulfil the following editorial requirements:

1) We note that the overall quality of the figures is below an acceptable production quality and that a few are in landscape rather than portrait orientation. Please improve them and make sure all figures are in portrait orientation. Please refer to our figure preparation guide for further information (embopress.org/sites/default/files/EMBOPress_Figure_Guidelines_061115.pdf).

2) We note that you have provided the EV legends and table in an Appendix style word document. Please move the legends to the main text and incorporate the supplementary materials & methods into the main manuscript section. Also, the EV tables should be uploaded individually with their legends. Please also update all manuscript callouts for the tables and figures to appropriate EV nomenclature: There are various references to Tables Sn, which instead should be Table EVn. The same goes for the EV figures, some of which are also still referred to as Fig. Sn. Please refer to our detailed author guidelines (embomolmed.embopress.org/authorguide#expandedview). Finally, the appropriate term is "expanded" view, not "extended" view.

3) Please choose 5 keywords only.

4) Multi-author references are currently listed as 10 authors et al. the correct format is 20 authors et al. Please correct.

5) As per our author guidelines, the description of all reported data that includes statistical testing must state the name of the statistical test used to generate error bars and P values, the number (n) of independent experiments underlying each data point (not replicate measures of one sample), and the actual P value for each test (not merely 'significant' or 'P < 0.05').

6) We encourage the publication of source data, with the aim of making primary data more accessible and transparent to the reader. Would you be willing to provide a PDF file per figure that contains the original, uncropped and unprocessed scans of all or at least the key gels used in the manuscript and/or source data sets for relevant graphs? The files should be labeled with the appropriate figure/panel number, and in the case of gels, should have molecular weight markers; further annotation may be useful but is not essential. The files will be published online with the article as supplementary "Source Data" files. If you have any questions regarding this just contact me.

7) Every published paper includes a 'Synopsis' to further enhance discoverability. Synopses are displayed on the journal webpage and are freely accessible to all readers. They include a short description as well as 2-5 one-sentence bullet points that summarise the key NEW findings of the paper. The bullet points should be designed to be complementary to the abstract - i.e. not repeat the same text. We encourage inclusion of key acronyms and quantitative information. Please use the passive voice. Please attach this information in a separate file or send them by email, we will incorporate it accordingly. We also encourage the provision of striking image or visual abstract to illustrate your article. If you do, please provide a jpeg file 550 px-wide x 400-px high.

Finally, while performing our standard pre-publication image check routines, we noted a few issues that require your action:

- 1) The first column in Fig 1B first appears to be duplicated/re-used in Fig. EV1B. Please explain
- 2) The central column in Fig. 1B appears to have completely empty panels, please explain.

Please submit your revised manuscript within two weeks. I look forward to seeing a revised form of your manuscript as soon as possible.

***** Reviewer's comments *****

Referee #2 (Remarks for Author):

The authors have significantly strengthened their report by including more samples, thus reducing the chance of reporting one-off results. A few minor suggestions:

1. Claims on finding something for the first time (bottom section "IDH mutant glioma display reduced energy potential") are unnecessary and in most cases hard to verify and/or incorrect. This is a general remark on the use of this type of wording and not specific to the data.
2. Energetic charge is reported to be statistically significantly different between IDHm and IDHwt glioma. These differences seem marginal with the highest IDHm exceeding the lowest IDHwt. This suggests that these data should be cautiously interpret and the claims softened. Could public expression datasets be mined to provide further evidence?
3. CBS expression is associated with prognosis in 1p/19q code1 patients, but the key sentence ("Interestingly, when investigating the relationship between CBS and patient prognostic...") refers to oligodendrogliomas. Please use consistent language on glioma subtypes. To relate high CBS expression to a potential for therapeutic targeting seems a huge leap that really requires further experimentation and I suggest softening the language.
4. Rather than using asteriks to indicate different levels of significance across the figures, why not simply show p-value i.e. fig 6? Or at least include the explanation of the asteriks in the figure rather than the written legend.

Referee #3 (Comments on Novelty/Model System):

I remain unconvinced about the model system and the choice of control. I acknowledge the difficulty in finding a better control when using the PDX model only. However, complementary approaches including mining of TCGA data (as suggested by another reviewer) and the use of xenograft models derived from IDH manipulated cells, could have been used to validate the PDX-derived data.

Referee #3 (Remarks for Author):

The resubmitted manuscript is improved because of the increase in the number of analyzed samples, the expanded statistical analysis and the partial modification of result interpretation and discussion. This reviewer remains unconvinced by the general experimental model and choice of control but acknowledges the difficulty of finding a more appropriate control when using the PDX model.

2nd Revision - authors' response

10 September 2017

Authors made requested editorial changes.

Corresponding Author Name: Simone P. Niclou

Manuscript Number: EMM-2017-07729